



# A systematic evaluation of high-cloud controlling factors

Sarah Wilson Kemsley[1*], Paulo Ceppi[2], Hendrik Andersen[3,4], Jan Cermak[3,4], Philip Stier[5], Peer Nowack[6]

[1] Climatic Research Unit, University of East Anglia, UK
[2] Department of Physics, Imperial College London, UK
[3] Institute of Meteorology and Climate Research, Karlsruhe Institute of Technology (KIT), Germany
[4] Institute of Photogrammetry and Remote Sensing, Karlsruhe Institute of Technology (KIT), Germany
[5] Atmospheric, Oceanic and Planetary Physics, Department of Physics, University of Oxford, UK
[6] Institute of Theoretical Informatics, Karlsruhe Institute of Technology (KIT), Germany

*Correspondence to*: s.wilson-kemsley@uea.ac.uk

**Abstract**

Clouds strongly modulate the top-of-the-atmosphere energy budget and are a major source of uncertainty in climate projections. "Cloud Controlling Factor" (CCF) analysis derives relationships between large-scale meteorological drivers and cloud-radiative anomalies, which can be used to constrain cloud feedback. However, the choice of meteorological CCFs is crucial for a meaningful constraint. While there is rich literature investigating ideal CCF setups for low-level clouds, there is a lack of analogous research explicitly targeting high clouds. Here, we use ridge regression to systematically evaluate the addition of five candidate CCFs to previously established core CCFs within large spatial domains to predict longwave high-cloud radiative anomalies: upper-tropospheric static stability ($S_{UT}$), sub-cloud moist static energy, convective available potential energy, convective inhibition, and upper-tropospheric wind shear. All combinations of tested CCFs predict historical, monthly variability well for most locations at grid-cell scales. Differences between configurations for predicting globally-aggregated radiative anomalies are more pronounced, where configurations including $S_{UT}$ outperform others. We show that for predicting local, historical anomalies, spatial domain size is more important than the selection of CCFs, finding an important discrepancy between optimal domain sizes for local and globally-aggregated radiative anomalies. Finally, we scientifically interpret the ridge regression coefficients, where we show that $S_{UT}$ captures physical drivers of known high-cloud feedbacks, and thus deduce that inclusion of $S_{UT}$ into observational constraint frameworks may reduce uncertainty associated with changes in anvil cloud amount as a function of climate change. Therefore, we highlight $S_{UT}$ as an important CCF for high clouds and longwave cloud feedback.



## 1 Introduction

Changes in clouds are the primary source of uncertainty in the quantification of equilibrium climate sensitivity (ECS) – the long-term global warming response to a doubling of atmospheric carbon dioxide (Sherwood et al., 2020; Zelinka et al., 2022). Cloud-induced radiative anomalies ($R$) at the top-of-the-atmosphere (TOA) refer to changes in the balance of incoming and outgoing radiation caused by interaction with clouds.

While most evidence suggests that the change in $R$ at the TOA as a function of global warming likely has a positive effect on Earth's energy balance and thus amplifies warming (e.g., Ceppi and Nowack, 2021), the magnitude of this global cloud feedback remains highly uncertain (Ceppi et al., 2017; Sherwood et al., 2020; Zelinka et al., 2022).

Motivated by the role of clouds as a key uncertainty factor, much progress has been made towards

understanding the mechanisms that drive changes in $R$, considering different cloud types under both natural unforced variability and long-term climate change. In particular, such work includes theoretical understanding of cloud feedback processes (e.g., Zelinka and Hartmann, 2010; Rieck, Nuijens and Stevens, 2012; Bony *et al.*, 2016); idealized regional modelling studies (Siebesma et al., 2003; Bretherton, 2015); convection-permitting global climate models (Rio et al., 2019); and climate model evaluation studies (Zelinka et al., 2022).


Here, we aim to systematically advance an alternative approach widely used for understanding and constraining uncertainties in cloud variability and trends in the form of Cloud Controlling Factor (CCF) analysis. Exploiting observed relationships between large-scale satellite cloud observations and meteorological predictor variables, CCF analyses have, for example, been used to derive observational constraints on cloud-related uncertainty estimates (Myers and Norris, 2016; Andersen et al., 2017, 2022; Fuchs, Cermak and Andersen, 2018;

Ceppi and Nowack, 2021; Myers et al., 2021). In particular, meteorological CCFs for low marine and boundary-layer clouds have been widely assessed (Qu et al., 2015; Brient and Schneider, 2016; Klein et al., 2017; Scott et al., 2020; Andersen et al., 2022), with typical frameworks including CCFs such as surface temperature ($T_{sfc}$), temperature advection, estimated boundary layer inversion strength (EIS), vertical velocity, 700 hPa relative humidity (RH) and near-surface wind speed. However, comparatively less research has specifically targeted the

CCFs for high clouds, despite their significant – and highly uncertain – contributions towards the total estimated feedback (Sherwood et al., 2020). A systematic comparison of CCF candidates for high clouds within a range of spatial domains will therefore be the main subject of this paper.



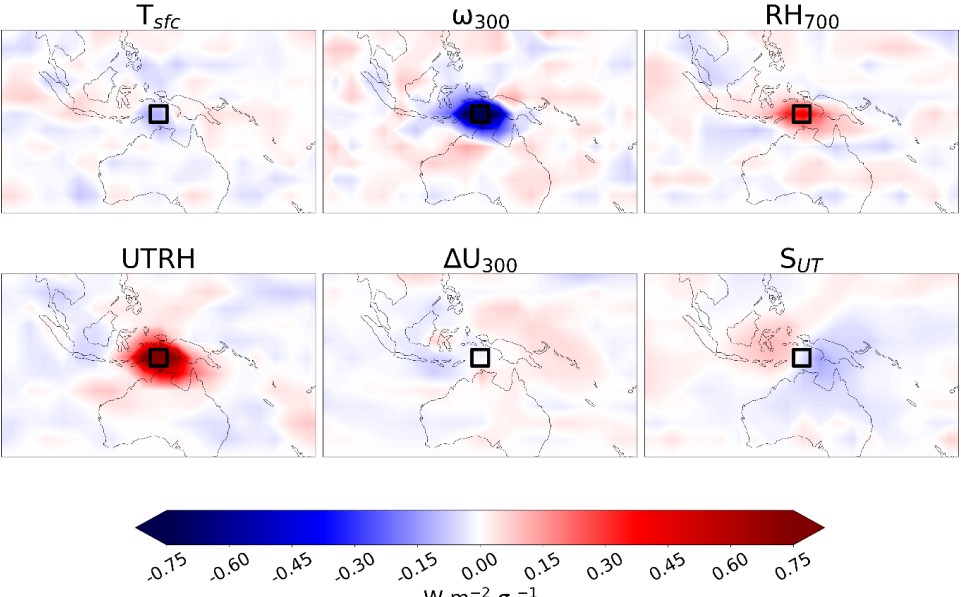

**Figure 1. CMIP multi-model mean longwave cloud-radiative sensitivities for a sample 5°× 5° target grid box (7.5° S, 132° E, indicated by the black box) to surface temperature ($T_{sfc}$), vertical velocity at 300 hPa ($\omega_{300}$), relative humidity at 700 hPa and in the upper troposphere ($RH_{700}$ and UTRH, respectively), wind shear at 300 hPa ($\Delta U_{300}$), and upper-tropospheric static stability ($S_{UT}$) using a 21x11 domain of grid-boxes around the target (corresponding to 110° longitude x 55° latitude area, centered on the grid-box.). Radiative anomalies are normalized for a one-SD (σ) anomaly in the controlling factors, based on monthly variability.**

Our work builds on a modification to a previous CCF approach, which was introduced by Ceppi and Nowack (2021, hereafter CN21). CN21 used ridge regression for their analyses, which allowed them to consider large spatial domains of CCF predictor patterns around target grid points in which cloud-radiative anomalies were predicted, with an example shown in Fig. 1. This approach contrasts with previous CCF analyses using standard multiple linear regression, which are constrained to a small number of predictors (typically < 10). This allowed their analysis to be extended beyond specific cloud regimes. As shown in CN21, the consideration of larger-scale CCF patterns led to improvements in predictive skill for both shortwave (SW) and longwave (LW) global cloud feedback. The intuition behind using spatial patterns of CCFs is motivated by the synoptic-scale atmospheric system within which the lifecycle of clouds – from formation to cessation – occurs, resulting in more robust predictions of global cloud feedback. Non-local features, such as large-scale patterns of sea-surface temperature anomalies and changes in the atmospheric circulation (e.g., convergence and divergence) are implicitly encoded using large spatial domains, which are not included in scalar CCF analysis despite their relevance for the context in which cloud development occurs (when considering monthly averaged data typically used for CCF analyses (Klein et al., 2017)). Altogether, considering larger-scale patterns resulted in better out-of-sample predictions,



which consequentially tightened the cloud-induced uncertainty in general circulation model (GCM)-modelled
ECS.

However, the framework introduced by CN21 highlighted an important limitation. As the same set of
five CCFs were used for SW and LW analyses, their predictive skill was markedly stronger for global SW and
net feedback components than for LW. Given that LW feedback is largely driven by high clouds, while SW
feedback is instead predominantly driven by the oft-studied low clouds, we speculate the performance deficit may
be – at least to a degree – a symptom of CCF choice. Indeed, Zelinka et al. (2022) specifically recommend that
the drivers of high cloud feedback must be targeted to reduce cloud-related uncertainty in ECS estimates.

To address these open questions, we use ridge regression to methodically assess candidate CCFs of high
clouds within a range of spatial scales, aiming to inform CCF choice for future observational constraints on the
ECS uncertainty. Here, we target LW cloud radiative anomalies ($R_{LW}$) as they are more directly associated with
high clouds than SW (and consequently, net) radiative anomalies. We briefly assess implications of CCF choices
on net anomalies, $R_{NET}$ noting that, historically, LW and SW high-cloud radiative anomalies tend to offset each
other, resulting in little net signal ($R_{NET}$) for thick clouds over monthly timescales.  We therefore restrict our
analysis to clouds with top pressures smaller than 680 hPa; future references to "$R$" are therefore specifically
emanating from these non-low clouds (see Sect. 3.1 for the dataset used). Though radiative effects from midlevel
clouds are also by definition included in our analysis, we collectively refer to radiative anomalies as "high"
henceforth for simplicity (Zelinka et al., 2016).

We systematically assess static stability in the upper troposphere ($S_{UT}$), sub-cloud moist static energy
($m$), convective available potential energy (CAPE), convective inhibition (CIN) and upper tropospheric wind
shear ($\Delta U$ for easterly shear) as CCFs in on the basis of their physical relationships with high-cloud properties or
convection, with an overview presented in Sect. 2. Aiming to inform choices for future *observational* constraint
analyses, we only suggest CCFs that are readily available (or easily calculated from measurable quantities).
Alternative variables, such as the radiatively-driven divergence, horizontal mass convergence, and gross moist
stability, may also capture high-cloud properties but their derivation requires numerical modelling and hence we
do not consider them here. Sections 3 and 4 discuss the data and methods we use, respectively, with combined
results and discussion presented in Sect. 5. We first discern which CCF combinations are able to best predict out-
of-sample grid-cell scale historical internal variability. We then investigate which combinations best predict out-
of-sample globally-aggregated $R_{LW}$. Based on the results of our statistical testing, we physically interpret the
coefficients for a single (deemed "optimal" by our analysis) configuration of CCFs, and assess whether the spatial
pattern, magnitude and variability of the cloud properties (i.e., cloud top pressure and cloud fraction) are
accurately captured.

**2 High-Cloud Controlling Factors**

Ubiquitously present over the tropics, cirrus, cirrostratus and deep convective clouds are responsible for
the largest annual-mean changes in global TOA LW flux (Chen et al., 2000). Tropical cirrus clouds develop
through one of two mechanisms: outflow from deep convective cores, or in-situ ice formation that is not associated
with convection (Gasparini et al., 2023; Kärcher, 2017). The former, referred to as "anvil cirrus" together with a
mature cumulonimbus core, form tropical anvil clouds. "Thick" cirrus are both effective absorbers of upwelling



LW radiation and also efficient reflectors of incident SW radiation. Over time, dynamical, radiative and microphysical processes can spread the thick anvil cirrus, extending anvil lifetime and resulting in larger cloud cover than the initial convective core (Luo and Rossow, 2004; Gasparini et al., 2023). Such processes can result in the formation of "thin" cirrus clouds, characterised by a relatively smaller SW cloud radiative forcing compared to LW (Jensen et al., 1994; McFarquhar et al., 2000). Though deep convective clouds presently have relatively small abundance (compared to other cloud types), their local radiative effects are large (Chen et al., 2000), and therefore changes to their frequency of occurrence can have substantial impacts on cloud feedback. Despite this, most previous CCF analyses focused on low-cloud regimes so that the selection and design of CCFs were mainly motivated by meteorological situations driving cloud formation and cessation in those cloud regimes (Klein et al., 2017).

In CN21, a compromise was sought by considering classic CCFs such as $T_{sfc}$, EIS, and $RH_{700}$ (relative humidity at 700 hPa), but by also using the vertical velocity at 500 hPa ($\omega_{500}$) and upper-tropospheric relative humidity (UTRH, the vertically averaged relative humidity in the 200 hPa layer below the tropopause) as predictors in an attempt to additionally target high clouds. In the following, we will build on the CN21 CCF set-up, specifically targeting modifications and additions that are more likely to represent state variables important for the aforementioned high clouds. One-by-one we will motivate these CCF candidates physically and formally define, then test the prediction results of possible CCF combinations for high clouds in Sect. 5.

An overview of all CCFs considered and their scientific motivations is summarised in Table 1. We keep $T_{sfc}$, $RH_{700}$, UTRH, and $\omega$ (at variable pressure levels) in all configurations, which we refer to as the "core" CCFs, as they jointly explain a large portion of historical variability in $R_{LW}$, and are each physically related to high-cloud formation. The large-scale distribution of tropical deep clouds is closely tied to the distribution of SSTs and upper-tropospheric relative humidity (Bony et al., 1997; Li et al., 2014), with research indicating that lower free-tropospheric relative humidity regulates the mean height of convective outflow (Sherwood et al., 2004). Vertical velocities ($\omega$) indicate regions of subsidence or ascent, with enhanced ascending motion supporting thicker, higher cloud layers (Ge et al., 2021). Andersen *et al.* (2023) find that the magnitude of (local) sensitivity to $\omega$ is largest at 300 hPa, hence we test vertical velocity at both 300 hPa and 500 hPa (used in CN21) in this study.

Estimated boundary-layer inversion strength (EIS) is *not* typically regarded a controlling factor for high clouds specifically, despite its wide use in general and low-cloud CCF analyses. This results in relatively little literature interpreting high-cloud sensitivities to EIS. Despite this, CN21 used *only* the $T_{sfc}$ and EIS sensitivities to observationally constrain global cloud feedback, for both SW and LW components. These sensitivities are suitably decoupled from the clouds, and still achieve good (albeit poorer than SW and net) predictions for global LW feedback. We therefore suggest five candidate CCFs as replacements for EIS that more directly represent convective processes or high-cloud formation, that are also sufficiently external to the clouds themselves and may be similarly used in constraints.

We list candidate CCFs (and EIS) and discuss them in turn below, with the exact definitions provided in Sect. 3.2:

- **Static stability** is the vertical gradient of potential temperature, measuring the stratification of the atmosphere (Grise et al., 2010). Upper-tropospheric static stability is robustly (negatively) correlated with upper-level cloud incidence over much of the global ocean (Li et al., 2014) and has been observationally linked with changes in tropical anvil cloud fraction through the "anvil



iris" thermodynamic mechanism (Bony et al., 2016; Saint-Lu et al., 2020, 2022). We expect increases in local upper tropospheric static stability to result in local reductions in high cloud fraction, with suppressed vertical motion;

- **Moist static energy** characterises the energy of an air parcel in a moist environment, considering its internal energy (latent and sensible heat) and potential energy due to its elevation. Sub-cloud moist static energy ($m$) may affect cloud formation, as higher levels of $m$ signify increased potential for uplift and condensation. Additionally, when buoyant air from the boundary layer fills the free troposphere, it can inhibit initiation of convection in colder regions,
setting a threshold that hinders further upward movement (Srinivasan and Smith, 1996; Zhang and Fueglistaler, 2020). We suggest that high $m$ increases local high-cloudiness, while in contrast, we speculate that non-local $m$ can either decrease (due to convective thresholds) or increase cloudiness (depending on horizontal transport);

- **Convective available potential energy (CAPE)** is a measure of deep instability, describing the
amount of energy available for an air parcel to rise freely through the atmosphere. CAPE offers insights into the onset, genesis and scale of atmospheric deep convection, and has been described as the fuel for a thunderstorm (Donner and Phillips, 2003; Jensen and Delgenio, 2006; Riemann-Campe et al., 2009). We speculate increased CAPE suggests an environment conducive to sustaining deep convection, and thus more high cloud;

- **Convective inhibition (CIN)**, a form of conditional instability and CAPE's opposing parameter, is a measure of the amount of energy required for a parcel to overcome a stable layer of air and initiate the development of deep convection. A large absolute value of CIN may indicate a stable atmosphere, and thus unfavourable conditions for the development of deep convective clouds (Louf et al., 2019). Note that high CIN is a required precursor for the buildup
of CAPE. Once CIN has been overcome, conditions are favourable for deep convection;

- **Wind shear,** defined here as the vertical change in horizontal wind speed, is an important dynamical characteristic of the upper troposphere. Wind shear influences the organisation of convective storms and mesoscale convective systems in various ways, though understanding its relationship with cloud properties has proved historically challenging (Anber et al., 2014).
However, studies suggest that wind shear can increase cloud-top turbulence, spread and stretch clouds horizontally through the advection of air at different levels and speeds, and hasten cirrus cloud dissipation (Lin and Mapes, 2004; Marsham and Dobbie, 2005; Jensen et al., 2011). We speculate wind shear mainly affects high-cloud fraction;

- **Estimated inversion strength (EIS)** describes the strength of the boundary layer and is a
dominant control for low-clouds (Wood and Bretherton, 2006; Andersen et al., 2022, 2023) and is widely used in general CCF analysis (CN21, (Klein et al., 2017). However, EIS is not considered a driver of high-cloud incidence, but CN21 speculated that EIS may function as a proxy for factors relating to deep convection.

Note that several candidate CCFs are not independent. For example, high values of CIN are required for
a buildup of CAPE, and a stable boundary layer may be represented by both high CIN and high EIS.



**Table 1. High-cloud controlling factors used in CN21 and proposed here, physical explanations connecting them to high clouds or convection, and the key studies supporting them. References to "clouds" in this table are for high clouds only. EIS is not a core CCF, and therefore for conciseness we include EIS under the "Candidate CCFs" subheading.**

| Cloud controlling factor | Physical explanation | Key studies |
|---|---|---|
| ***Core cloud controlling factors*** | | |
| Surface temperature ($T_{sfc}$) | Warming surface temperature heats atmospheric column; large-scale distribution of clouds is tied to atmospheric profile of temperature; anvil clouds approximately rise with isotherms. | (Bony et al., 1997; Zelinka and Hartmann, 2011) |
| Free-tropospheric relative humidity ($RH_{700}$) | Regulates mean height of convective outflow. | (Sherwood et al., 2004) |
| Upper-tropospheric relative humidity (UTRH) | Tropical clouds tied to spatial distribution of UTRH and lifetime of anvil clouds. A reciprocal relationship may exist; UTRH modulated by detrainment. | (Bony et al., 1997; Li et al., 2014) |
| Vertical pressure velocity ($\omega$) | Indicates regions of ascent and subsidence. Enhanced ascending motion supports thicker clouds. | (Ge et al., 2021) |
| ***Candidate CCFs*** | | |
| Estimated boundary layer inversion strength (EIS) | Limited literature; perhaps serves as a proxy for deep convective processes; strength of boundary layer inhibits convection. | CN21 |
| Upper-tropospheric static stability ($S_{UT}$) | Static stability associated with radiatively driven convergence; anvil altitude and amount collocate with peak convergence. | (Zelinka and Hartmann, 2010; Li et al., 2014; Bony et al., 2016; Saint-Lu et al., 2020, 2022) |
| Convective Available Potential Energy (CAPE) | Measure of deep instability; indicates energy available for convection. | (Donner and Phillips, 2003; Jensen and Delgenio, 2006; Chakraborty et al., 2016; Louf et al., 2019) |
| Convective Inhibition (CIN) | Shallow instability; indicates the energy required to leave stable boundary layer. | (Louf et al., 2019) |
| Sub-cloud moist static energy ($m$) | Moisture content of sub-cloud atmosphere fuels convection. | (Zhang and Fueglistaler, 2020) |
| Upper-tropospheric wind shear ($\Delta U_{300}$) | Influences organisation of convective storms; affects cloud-top turbulence and mesoscale anvil formation; affects cloud cover. | (Lin and Mapes, 2004; Marsham and Dobbie, 2005; Jensen et al., 2011). |

**3 Data**

We use monthly-mean (unless explicitly mentioned otherwise) cloud property and CCF data, re-gridded to a common 5°x5° resolution. At these spatial and temporal scales, we expect the clouds to be approximately in equilibrium with their environment (Klein et al., 2017). To represent observed cloud-radiative data, we use combined Moderate Resolution Imaging Spectroradiometer (MODIS) retrievals from both Aqua and Terra



instructions... instruments, identified as MCD06COSP (Pincus et al., 2023). These retrievals are included as part of the CFMIP Observation Simulator Package (COSP, where CFMIP refers to the Cloud Feedback Model Intercomparison Project), which facilitates the evaluation of models against observations in a consistent manner (Bodas-Salcedo et al., 2011). For climate model data, we use eighteen GCMs that have run the International Satellite Cloud Climatology Project (ISCCP) simulator (Zelinka et al., 2012a) from the Coupled Model Intercomparison Project phases 5 and 6 (CMIP5/6). For a full list of CMIP models used in this research, see Supplementary Material Sect. S1. For the meteorological CCFs we use ERA5 reanalysis data at monthly resolution, with the exception of CAPE and CIN which we first calculate using *daily* air temperature and relative humidity profiles, and then take the monthly mean. We use reanalysis data as a proxy for direct observations; henceforth, when "observed" results are discussed, we refer to predictions made for observed radiative anomalies using ERA5 meteorological CCFs.

We restrict the CMIP datasets to twenty years, aligned with the length of available observational record, though with slightly different time periods. For observations, data is available from July 2002 to June 2022. For the CMIP models, we use historical data from January 1981 to December 2000. We use this period because it is close to the present-day climate, under the constraint of availability of historical CMIP data (and noting that only a small set of models provide satellite simulator output for the RCP and SSP scenarios). For predictions of observed and modelled $R_{LW}$, we restrict our analysis from 60°S – 60°N. As is commonplace in CCF analysis, the seasonal cycles (climatological averages of each month) have been removed from the CCFs and radiative anomalies (Myers et al., 2021; Andersen et al., 2022). Prior to analysis, predictor variables are scaled to unit variance and zero mean to weight signals equally in the optimisation process (Scott *et al.*, 2020, CN21).

## 3.1 Cloud property histograms

Our analysis is based on histograms of cloud fraction as a joint function of cloud top pressure (CTP) and cloud optical depth ($\tau$). Cloud-radiative kernels are used to convert binned cloud amount anomalies into top-of-atmosphere radiative flux anomalies, and to partition these into contributions from changes in cloud top pressure ($CTP$), cloud fraction ($CF$), and optical depth ($\tau$), with a small residual contribution (Zelinka et al., 2012a, b, 2016). The cloud-radiative kernels we use here were first introduced in Zelinka et al. (2012a), with an improved decomposition method presented in Zelinka et al. (2016). Note that the same kernels (developed using ERA5 Interim temperature, humidity and ozone profiles) are used to decompose both the observed and modelled radiative anomalies. Cloud-radiative kernels are available from https://github.com/mzelinka/cloud-radiative-kernels.

## 3.2 Meteorological cloud controlling factors

Static stability is calculated using monthly air temperature ($T_p$) at CMIP standard pressure levels, $p$,

$$S_p = \frac{R_C T_p}{C p} - \frac{dT}{dp} \tag{1}$$

where $S_p$ is the static stability at pressure $p$, $C$ is the specific heat at constant pressure, and $R_C$ the gas constant. We define upper tropospheric static stability, $S_{UT}$, at the standard pressure level closest to the tropopause height plus 50 hPa in pressure units, where the monthly-mean tropopause is calculated using the standard WMO definition (Reichler et al., 2003). This is to ensure that our definition of $S_{UT}$ accounts for the zonal distribution of tropopause height.



Moist static energy, CAPE and CIN are calculated using the Metpy V1.3.1 Python package (May and Bruick, 2019). Moist static energy is calculated at standard pressure levels using monthly air temperature and relative humidity datasets. To approximate sub-cloud moist static energy, $m$, we average moist static energy from the surface to (and including) 700 hPa. We use MetPy's "most unstable" CAPE and CIN function, which we calculate for all available CMIP models and ERA5. This involves calculating the most unstable air parcel from the temperature and humidity profiles, and hence calculating CAPE and CIN using this parcel. CAPE and CIN are first calculated using daily temperature, humidity and pressure values at standard CMIP pressure levels and then averaged for each month. Of the eighteen CMIP models, daily datasets for atmospheric temperature and humidity are only available for fourteen of the models (see Sect. S1 in Supplementary Material).

Free-tropospheric vertical wind shear is calculated as the difference in 925 hPa and 300 hPa easterly wind speeds, $U$, standardised by the change in geopotential height, $z$, where

$$\Delta U_{300} = \frac{U_{300} - U_{925}}{z_{300} - z_{925}} \qquad (2)$$

with subscripts referring to the pressure levels for each variable (Chakraborty et al., 2016). Both easterly and northerly wind shear have been assessed, though we only discuss easterly shear here as overall performance metrics are relatively consistent between the directions of shear.

$T_{sfc}$, $\omega_{300}$, $\omega_{500}$ and $RH_{700}$ are directly observable or modelled quantities. We define EIS and UTRH consistently with CN21. EIS is a measure of lower-tropospheric stability, defined relative to the temperature-dependent moist adiabatic lapse rate (Wood and Bretherton, 2006) over global oceans. Over land, this is simply defined as the difference between the potential temperature at 700 hPa and the surface (Klein and Hartmann, 1993). UTRH is the vertically averaged relative humidity within the 200 hPa-layer below the tropopause (again defined using the WMO standard definition). Monthly-mean climatologies for all CCFs can be found in Fig. S1.

## 4 Method

### 4.1 Ridge regression

We use ridge regression to estimate sensitivities of cloud-radiative anomalies to changes in surrounding meteorological CCFs within two-dimensional spatial domains. While still being a linear least-squares regression approach, the inclusion of an L2-regularization penalty term means that the method can more effectively deal with high-dimensional regression problems than unregularized multiple linear regression (Hoerl and Kennard, 1970; CN21; Nowack et al., 2021). This, in turn, allows us to consider larger domains of CCFs as predictors in the first place, leading to improved generalized predictive skill. The spatial domain within which CCFs are used to predict $R$ at a central grid-cell, $r$, is referred to by the number of grid-cells in a longitude x latitude space (i.e., a 7x3 domain corresponds to 35° longitude x 15° latitude, see also Fig. 1). Five domain sizes are tested: 1x1, 7x3, 11x5, 15x9 and 21x11.

Statistical cross-validation is used to optimise the regression fit by minimising the cost function,

$$J_{ridge} = \sum_{t=1}^{n} \left( R(r)_t - \sum_{i=1}^{M} c_i X_{i,t} \right)^2 + \alpha \sum_{i=1}^{M} c_i^2 \qquad (3)$$



which puts a penalty on overly large regression coefficients, $c_i$; where $n$ is the number of datapoints; $X_{i,t}$ is the $i$-th CCF at time $t$; $M$ is the number of dimensions in the model (i.e., for a 7x3 domain using five unique CCFs, $M = 7\,x\,3\,x\,5 = 105$); and α the regularisation parameter.

The first term in Eq. (3) is the ordinary least squares regression error. We classically approximate $R(r)$ by a linear function of anomalies in the set of $M$ cloud controlling factors,

$$R(r) \approx \sum_{i=1}^{M} \frac{\partial R(r)}{\partial X_i}\, \partial X_i. \tag{4}$$

We refer to

$$\Theta_i(r) = \frac{\partial R(r)}{\partial X_i} \tag{5}$$

as the sensitivities, $\Theta_i(r)$, of $R(r)$ to anomalies in the $i$-th CCF. Again, see Fig. 1 for an example of the spatial pattern of for six CCFs using a 21x11 domain.

Using fivefold cross-validation, we determine the optimal value for the regularization parameter, α, where the second term in Eq. (3) is the L2-regularization penalty. We split the timeseries into five ordered time slices and optimise α by fitting Eq. (3) to each of four slices at a time. Optimal α is hence found by evaluating predictions on the fifth time slice using the $R^2$ score independently for each location in the observed and modelled datasets.

For Sect. 5.1, 5.2 and 5.4 we use sensitivities to subsequently predict two years of withheld data. We rotate the withheld dataset every two years, resulting in ten unique training-validation dataset combinations. Predictions are subsequently concatenated, resulting in a continuous twenty-year timeseries predicted "out-of-sample", with no datapoint having been predicted using the same dataset that the model was trained on. Standard performance metrics (Pearson $r$ correlation coefficient, $R^2$ score, and root mean squared error, RMSE) are calculated using the concatenated predictions and the original twenty-year dataset. For Sect. 5.3, we use the sensitivities estimated from the full twenty-year dataset to visualise spatial distributions.

**5 Results and Discussion**

Here we present results for the CCF analyses for $R_{LW}$, including a systematic assessment and intercomparison of possible CCF configurations. "CCF configuration" refers to the combination of meteorological variables used to predict $R_{LW}$. Configurations are labelled based on which of the proposed CCFs (shown in Table 1) are used in addition to the following core retained factors $T_{sfc}$, $\omega_{300}$, $RH_{700}$, and UTRH (i.e., configuration $S_{UT}$ refers to predictions made using $T_{sfc}$, $\omega_{300}$, $RH_{700}$, UTRH and $S_{UT}$ ). Where appropriate, we additionally point to the corresponding $R_{NET}$ results in the Supplementary Material.

In the following, we compare CCF configurations using standard performance metrics for time series predictions. Since we learn separate CCF functions to predict $R_{LW}$ at each 5° x 5° grid-point, we first evaluate the prediction performance of those functions individually, which we refer to as "local" predictions. We then average local performance metrics near-globally (i.e., for all available predictions, 60°S – 60°N inclusive), henceforth simply referred to as "globally" averaged, with grid-cells weighted by the cosine of their latitude. We also average metrics in the tropical ascent regions, which we define as grid-cells with observed climatological EIS < 1 K, $\omega_{500}$ < 0 hPa s$^{-1}$, and latitude equatorward of 30° (Medeiros and Stevens, 2011).



We note that not all regions are equally cloudy, which leads to differently high levels of variance in $R_{LW}$ and, thus, signals for the regression to learn from. As a result, performance metrics tend to be lower in subsidence regions where there are few high clouds. For the global performance averages, we have addressed this issue by

weighting grid-cells based upon their climatological mean $R_{LW}$ in addition to the cosine of their latitude, to avoid penalising average metrics from low scores in regions with relatively little signal.

Using the CCF framework, an observational constraint on global cloud feedback can be made using local $R_{LW}$ predictions under a forcing (such as $4xCO_2$) that are aggregated globally and normalised by the change in global mean surface temperature. Though we do not predict feedback here, we instead assess which CCF

configuration best estimates the globally-aggregate $R_{LW}$ by spatially averaging each local prediction and target value globally (and in tropical ascent regions) first, and then calculating the performance metrics. Henceforth, note a distinction between globally averaged *metrics* for local predictions (e.g., Fig. 2a-b) and metrics for globally-*aggregated $R_{LW}$* (e.g., Fig. 2c-d).

### 5.1 Predictive skill on observations

We first assess CCF configuration skill for local predictions, with results shown in Fig. 2a-b (with columns c-d showing globally-aggregated results). Using ridge regression, all configurations predict out-of-sample local $R_{LW}$ well. This is reflected by local predictions that are strongly correlated to the observed values, with Pearson $r$ exceeding 0.75 and $R^2$ scores larger than 0.60, at all domain sizes (shown in Fig. S2). To demonstrate the strengths of ridge regression while using collinear predictors in high dimensions, we briefly

compare our results to the traditional multiple linear regression (MLR) approach. Using a 1x1 domain, there is little difference in skill between predictions made with MLR and ridge regression. However, while the predictive skill of ridge regression improves with increasing domain size and thus, the number of dimensions in the model, the opposite is true for MLR. Beyond 7x3, MLR coefficients become unstable, resulting in increasingly poor performance, and therefore these results are not shown.

We find local performance to depend both on the CCF configuration, with $EIS$ $(\omega_{500})$ exhibiting the weakest performance, and, more prominently, on domain size (Fig. 2a-b) (note that $EIS$ $(\omega_{500})$ is the configuration used in CN21). Correlation matrices are qualitatively consistent between the performance metrics, though improvements are more pronounced for RMSE (at local scales) than for Pearson $r$ and $R^2$ (likely owing to strict upper and lower limits for these metrics). We therefore show the RMSE in Fig. 2, with $R^2$ and Pearson $r$

shown in Fig. S2. Though changes in local skill (when globally averaged) between the configurations are subtle,





we reaffirm that they are indeed robust in Sect. 5.2, showing qualitatively consistent results for the CMIP models. In this section, we will discuss configurations for the optimal domain size for local predictions (7x3).

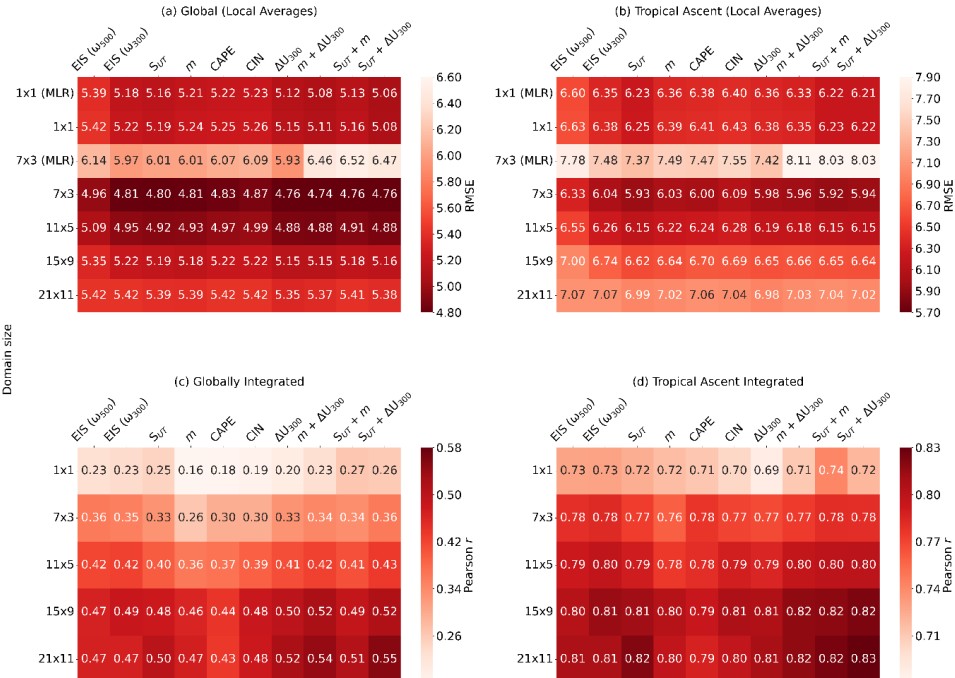

**Figure 2. Matrices showing skill metrics for predictions made for the observed R$_{LW}$ time series at each domain size using different "CCF configurations". A "CCF configuration" refers to the selection of cloud controlling factors used to predict R$_{LW}$. Each configuration uses T$_{sfc}$, RH$_{700}$, UTRH and ω$_{300}$ (with the exception of the first column, where ω$_{500}$ is used instead) and a candidate CCF(s) (e.g., S$_{UT}$), which is used to label each column. Predictions are made locally, with the RMSE averaged (a) globally and (b) in tropical ascent regions defined as grid-cells with observed climatological EIS < 1 K, ω$_{500}$ < 0 hPa s$^{-1}$. Metrics are weighted by the cosine of latitude and monthly standard deviation of R$_{LW}$ of each grid-cell. Pearson r is also shown for aggregated predictions, (c) globally and (d) in the tropical ascent regions, and compared to similarly aggregated observations. All predictions are made using ridge regression, except for row 1x1 (MLR) and 7x3 (MLR) in panels (a) and (b), which are made using multiple linear regression. Note different scales for each colorbar, with darker regions indicating higher skill (lower RMSE, higher Pearson r).**

In line with Andersen et al. (2023), we find the largest improvement in $R_{LW}$ predictive skill is achieved through changing ω from 500 hPa to 300 hPa (columns 1 and 2 of Fig. 2a-b). This suggests ω$_{300}$ is indeed an important predictor for deep convective and cirrus cloud radiative effects (Ge et al., 2021). We do, however, find that this results in a slight drop in performance for $R_{NET}$ (Fig. S3). We speculate that this is because 500 hPa instead better targets midlevel clouds which drive a shortwave contribution to $R_{NET}$ that is not present for $R_{LW}$. However, comparing across configurations using the same vertical velocity reveals that the best performing configurations for $R_{NET}$ generally align with $R_{LW}$ and owing to its strong longwave performance, for configurations henceforth we choose to replace ω$_{500}$ with ω$_{300}$.

When globally averaged, changes to the local performance metrics beyond changing the vertical pressure velocity are small (Fig. 2a-b, and Fig. S2). This is likely because a large proportion of the monthly variability is already explained using only T$_{sfc}$, ω$_{300}$, RH$_{700}$ and UTRH without the inclusion of additional CCFs (i.e., for 7x3, R$^2$ = 0.64 using core CCFs, compared with R$^2$ = 0.69 using $EIS$ ($w_{300}$) in addition). In the tropical ascent regions, improvements are highest using $S_{UT} + m$, though more generally any configuration using $S_{UT}$. This is reflected






by the spatial distribution of the performance metrics, where any configuration including $S_{UT}$ sees improvements to the skill metrics over the tropics. We additionally find more prevalent improvements in the extratropics using $\Delta U_{300}$ (spatial distributions shown in Fig. S4).

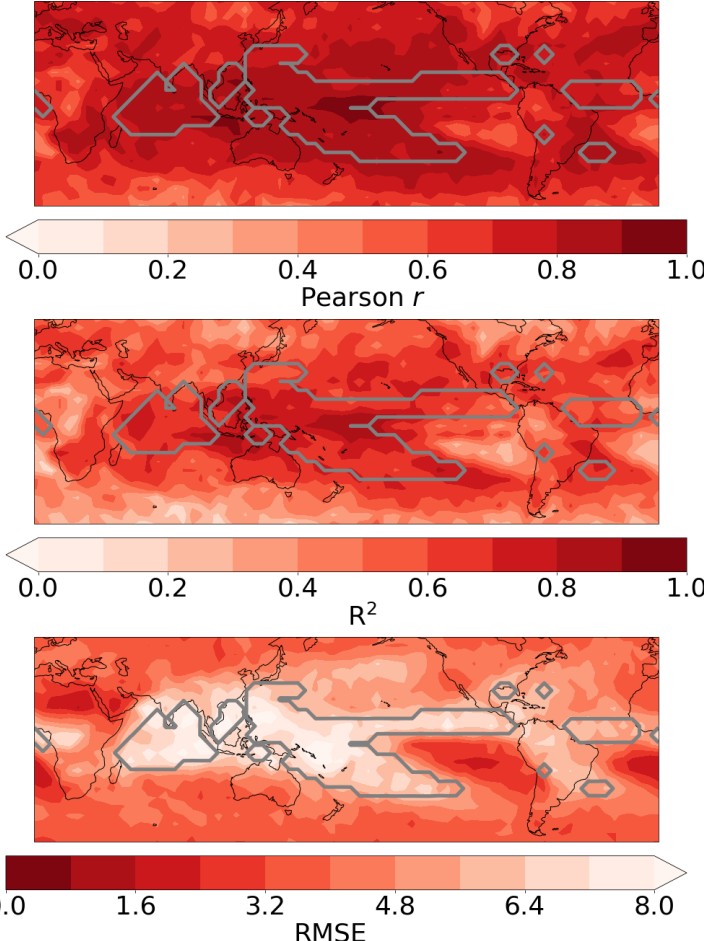

**Figure 3. Skill metrics (from top to bottom: Pearson $r$, $R^2$ score, and RMSE) for local predictions of observed $R_{LW}$ time series using the configuration $S_{UT} + \Delta U_{300}$ (with T$_{sfc}$, RH$_{700}$, UTRH and ω$_{300}$), with CCFs within a spatial domain of 21x11. Grey contours show the tropical ascent regions, defined as grid-cells with observed climatological EIS < 1 K, ω$_{500}$ < 0 hPa s$^{-1}$, and latitude equatorward of 30° (Medeiros and Stevens, 2011). Note different scales for each colorbar, with dark regions showing "better" skill metrics.**

Figure 3 shows an example of the performance metrics' spatial distributions for the configuration $S_{UT}$ +
$\Delta U_{300}$. The $r$ and $R^2$ are largest in the tropics where high clouds are ubiquitous and there is a large standard deviation in monthly $R_{LW}$, resulting in a strong signal for the regression model to learn from. Lower scores are present in the subsidence regions and the Southern Ocean. Despite low $R^2$ scores in subsidence regions, the Pearson $r$ is generally high. By contrast, the RMSE is typically much smaller here than in the tropical ascent regions, where local $R^2$ approaches unity. At first this may seem counterintuitive; however, the magnitude of the
$R_{LW}$ signal in the tropical ascent regions is much larger than in subsidence regions. This means that, although a





greater proportion of the tropical ascent monthly variability is captured (represented by the high $R^2$), the absolute difference between the observed and predicted signals may be larger than in subsidence regions where the RMSE is constrained by the small signal. This suggests that it is the small $R_{LW}$ signal that is predominantly responsible for the low $R^2$ scores and not poor model setup. In the Southern Ocean, we additionally speculate that poorer

performance may be attributable to a reduced quality of reanalysis data, arising from fewer observations available for assimilation (Mallet et al., 2023).

Neither CAPE nor CIN improve local predictive skill when globally averaged compared to alternative candidate CCFs. This appears partially due to being particularly poor predictors in the high-latitude extratropics. This is unsurprising; CAPE and CIN have been included as a CCF for their links to deep convection, which is not

frequent outside of the warm tropics. Additionally, literature hints at a potentially non-linear relationship between CAPE, CIN and high-cloudiness that would not be captured by the linear ridge regression. For example, in high-CAPE environments it is thought that there may generally be enough CAPE for convection to occur, indicating that the exact magnitude of CAPE is less important than passing a threshold for the onset of deep convection (Sherwood, 1999).

We now assess predictive performance on the globally-*aggregated* $R_{LW}$ time-series, with results shown in Fig. 2c-d. Here we use the Pearson $r$ to determine whether global trends are captured, though we again find that correlation matrices for each skill metric show qualitatively similar results (not shown). We again find improved skill for most of the CCF configurations compared to EIS ($\omega_{500}$) (i.e., the CN21 configuration), with performance also dependent on domain size for $R_{LW}$ (Fig. 2c-d). While local prediction performance peaks at 7x3,

we find a discrepancy with the globally-aggregated performance, which instead *increases* with domain size. For some configurations, $r$ continues to increase beyond 21x11, though this begins to tail off (not shown). The globally-aggregated results now align with the findings of CN21, where they show that the correlation between observed and predicted global cloud *feedback* increases with domain size. However, as domain size increases, so too do the number of dimensions in the model and thus the complexity. Owing to the trade-off between small

improvements and increased complexity, we restrict our analysis to 21x11 and below, and henceforth discuss results using the 21x11 domain.

We find that the performance metrics for globally-aggregated $R_{LW}$ are comparatively worse than the globally-averaged *local* metrics. We suggest that this may be caused by accumulation of local errors, in addition to weaker variability in the globally-aggregated $R_{LW}$ anomalies. However, while local predictions show only

modest improvements between CCF configurations, we find more marked differences for the global predictions. Changing the pressure level of $\omega$ no longer results in the largest – if any – improvements. Instead, the addition of $S_{UT} + \Delta U_{300}$ sees the largest leap in performance for global, and tropical ascent, predictions. This is generally consistent for $R_{NET}$, where $S_{UT}$ configurations outperform other candidate CCFs (see Fig. S3c-d).

### 5.1.1 CCF importance at different spatial scales

We suggest the discrepancy between optimal configurations may partially be caused by the relative significance of each CCF varying at different spatial scales. Owing to the linearity of ridge regression, we can partition the predicted local $R_{LW}$ signal into contributions from each CCF, such that (for example)

$$R_{LW} = R_{LW(T_{sfc})} + R_{LW(RH_{700})} + \cdots + R_{LW(\Delta U_{300})},$$  (6)



where $R_{LW(T_{sfc})}$ is the component of $R_{LW}$ predicted using only $T_{sfc}$ within the specified domain size (here we use

21x11), and so on for each CCF in the configuration. For each CCF, we calculate the explained variance fraction

(EVF) for $R_{LW}$ at each grid-cell. Equation (6) can similarly be repeated for the global $R_{LW}$ predictions, where local

predictions are first globally-aggregated for each CCF and then summed.

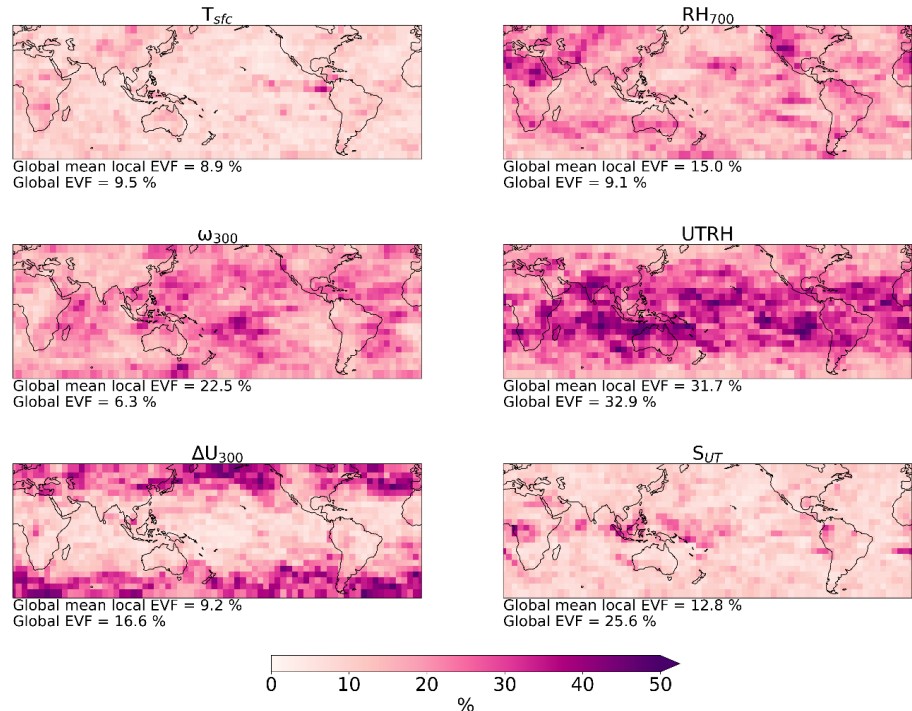

**Figure. 4. Maps showing the explained variance fraction (EVF) as a percentage for local predictions of $R_{LW}$ using a 21x11 domain and using configuration $S + \Delta U_{300}$ (with $T_{sfc}$, $RH_{700}$, UTRH and $\omega_{300}$). "Global mean local EVF" refers to the global mean EVF from local predictions, weighted by the cosine of each grid-cell's latitude. "Global EVF" refers to the EVF for each CCF's contribution to the globally-aggregated $R_{LW}$.**

We find that the relative importance of the CCFs varies depending on whether predictions are assessed

locally or globally-aggregated. Note that UTRH is consistently the most "important" predictor at both local and

globally-aggregated scales. It is plausible that there is bidirectional causality, where the presence of high cloud

influences UTRH by modulating the moisture content in the upper troposphere (i.e., outflow from convective

anvils).

Several studies point to thermodynamic changes dominating over dynamical effects for globally-

aggregated cloud feedback, likely because dynamical effects cancel out at sufficiently large scales (Bony et al.,

2004; Xu and Cheng, 2016; Byrne and Schneider, 2018). Conversely, thermodynamic and dynamical feedbacks

have more comparable importance at more local scales. We find our results are broadly analogous to this. For

configuration $S_{UT} + \Delta U_{300}$, $\omega_{300}$ is the second most "important" predictor, contributing over 20 % of the

variability in local $R_{LW}$ predictions, shown in Fig. 4. However, the EVF for $\omega_{300}$ substantially decreases when

globally-integrating $R_{LW}$ to only 6.3 %. This additionally explains why there is little difference between the

performance of $\omega_{300}$ and $\omega_{500}$ in Fig. 2c-d, despite resulting in the largest improvement (for a single CCF change)





at local scales. On the contrary, we find that $S_{UT}$ accounts for only 12.8 % of the local EVF (when globally averaged), making it the fourth (out of six) most "important" predictor. For global $R_{LW}$ predictions, $S_{UT}$ instead accounts for 25.6 % of the EVF.

## 5.2 Predictive skill on CMIP models

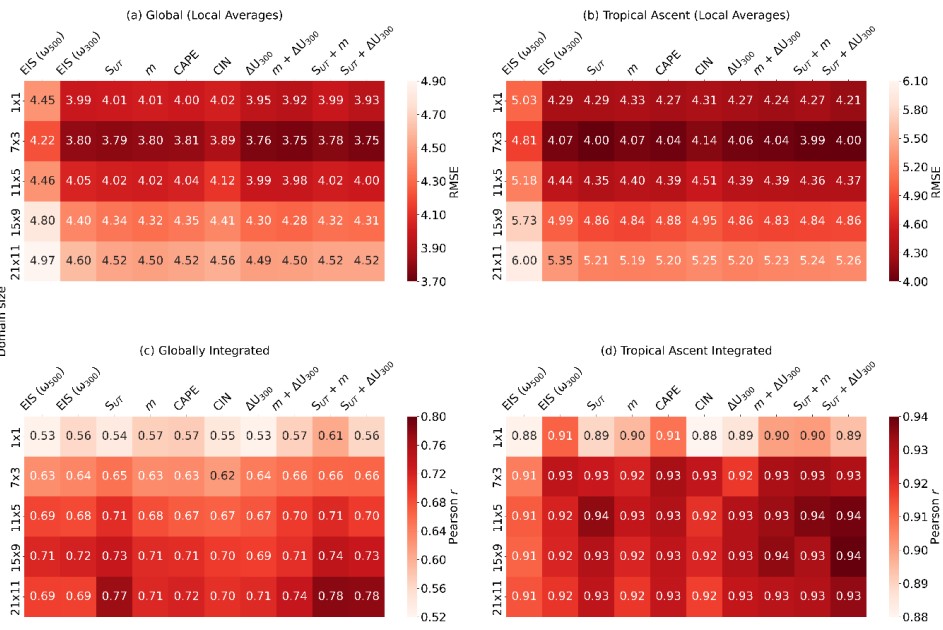


**Figure 5.** Matrices showing the Pearson $r$ score between the observed $R_{LW}$ time series and predictions made at each domain size using different "CCF configurations". A "CCF configuration" refers to the selection of cloud controlling factors used to predict $R_{LW}$. Each configuration uses $T_{sfc}$, $RH_{700}$, $UTRH$ and $\omega_{300}$ (with the exception of the first column, where $\omega_{500}$ is used instead) and a candidate CCF(s) (e.g., $S_{UT}$), which is used to label each column. The median has been calculated from 14 of the CMIP models (excluding the additional 4 without CAPE or CIN). Predictions are made locally, with the RMSE averaged (a) globally and (b) in tropical ascent regions, defined as grid-cells with observed climatological EIS < 1 K, $\omega_{500}$ < 0 hPa s$^{-1}$. RMSE is weighted by the cosine of latitude and monthly standard deviation of $R_{LW}$ of each grid-cell. Predictions are hence aggregated (c) globally and (d) in the tropical ascent regions and compared to similarly aggregated observations using Pearson $r$. Here, all predictions are made using ridge regression. Note different scales for each colorbar, with darker regions indicating higher skill (lower RMSE, higher Pearson $r$).

Now, we briefly present results for the CCF configurations using the CMIP5/6 models. Key questions are whether the CCF approach performs similarly between models and observations, and if there are any obvious discrepancies that could point towards the analysis framework being less applicable than in observations. Performance metrics are first calculated locally for each GCM. Independently for each GCM, local metrics are

meaned globally and in tropical ascent regions. The multi-model median result is then taken, with results shown in Fig. 5a-b. Finally, we integrate predictions globally, and in tropical ascent regions, independently for each GCM. The predicted global and tropical ascent-aggregated time series are compared against the similarly aggregated target values, with multi-model medians shown in Fig. 5c-d. Again, note a distinction between globally averaged, local performance *metrics*, and globally-*aggregated $R_{LW}$*.



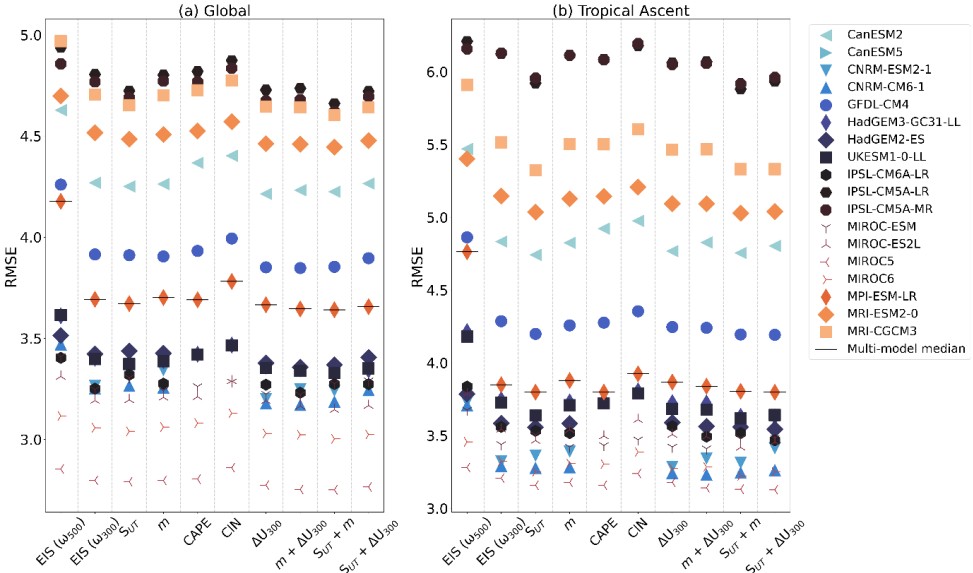

**Figure 6. Individual model mean RMSE for local predictions made at the optimal 7x3 domain size using different "CCF configurations". A "CCF configuration" refers to the selection of cloud controlling factors used to predict $R_{LW}$. Each configuration uses $T_{sfc}$, $RH_{700}$, UTRH and $\omega_{300}$ (with the exception of the first column, where $\omega_{500}$ is used instead) and a candidate CCF(s) (e.g., $S_{UT}$), averaged (a) globally and (b) in the tropical ascent regions, with RMSE weighted by each grid-cell's latitude and standard deviation for each of the 18 CMIP models. The multi-model median RMSE is shown from the 14 CMIP models that have CAPE and CIN available. Note that smaller values of RMSE correspond to a "better" configuration, and RMSE is higher in the tropical ascent regions owing to larger signals (see Fig. S5).**

CMIP results for local performance metrics (Fig. 5a-b) are largely analogous to the observations, with raising the pressure level of the vertical velocity resulting in the largest (though modest) improvement for $R_{LW}$ (but similarly decreasing skill for $R_{NET}$) and an optimal domain of 7x3. Multi-model median Pearson $r$ and $R^2$ metrics are higher than in the observations, which may be expected due to intrinsic knowledge of the meteorological conditions embedded within the CMIP models. Slightly suppressed metrics for observed $R_{LW}$

could additionally be caused by a potential mismatch between the observed radiative anomalies and the reanalysis meteorological variables. This therefore results in metrics that are more consistent between CCF configurations than for the observed $R_{LW}$. This is possibly due to the lower (higher) RMSEs (Pearson $r$ and $R^2$ scores) in the first place, which leaves less room for improvement. This is noticeable in Fig. 6, where models with poorer skill metrics (e.g., IPSL-CM5A-MR) show larger differences between CCF configurations than those with higher scores (e.g.,

MIROC5). Regardless, configurations $m + \Delta U_{300}$ and $S_{UT} + \Delta U_{300}$ show the largest improvements to the predictive skill globally, and configurations containing $S_{UT}$ in the tropical ascent regions.

       Highlighting the uncertainties within the CMIP models themselves, there is a large spread in the global-mean skill metrics shown in Fig. 6. For example, at the 7x3 domain, configuration $S_{UT} + \Delta U_{300}$ spans global-mean RMSE scores from 2.75 to 4.66 (and Pearson $r$ from 0.85 to 0.92, not shown). In the tropical ascent regions,

there is larger spread in the skill metrics, with IPSL-CM5A-MR and IPSL-CM5A-LR highlighted as upper outliers. There is also a slight dependency on configuration regarding the inter-model spread of predictive skill, where we find that $S_{UT}$ results in the lowest inter-model spread, and the upper outliers with the lowest RMSEs in tropical ascent regions.



The spatial distributions of the performance metrics are qualitatively very similar to the observations,
with the highest $r$ and $R^2$ in the tropical ascent regions, and lowest over the Southern Ocean and subsidence
regimes (see Fig. S5). While much of the Southern Ocean $R^2$ fell between 0.2 and 0.3 for the observations, the
multi-model median $R^2$ generally exceeds 0.5 (though again, with outliers). This helps to support our attribution
of lower predictive skill over the Southern Ocean in the observations, at least in part, to the known lower quality
of reanalysis datasets in this region (Mallet et al., 2023).

For predictions of globally-aggregated $R_{LW}$, skill metrics are again higher for the CMIP models than for
the observations, though performance similarly peaks at 21x11, and the general pattern of the correlation matrices
shown in Fig. 5c-d is similar to Fig. 2c-d. Here, any configuration including $S_{UT}$ predicts global $R_{LW}$ with the
highest correlation coefficient. For the CMIP results, CAPE performs comparatively better than the observations.
We speculate that the improved performance of CAPE in the models relative to the observations may be due to
the way in which convection is parameterised in models. This would thus result in stronger modelled relationships
between cloud-radiative anomalies and CAPE than exist in the observations.

Once again, the correlation matrices for $R_{NET}$ generally follow $R_{LW}$, though configuration $S_{UT}$ (without
an additional sixth CCF) performs best, with $\Delta U_{300}$ performing worse at every domain size (Fig. S3c-d). We
speculate that shear mostly reflects dynamically-driven cloud anomalies that compensate each other at large
scales, and hence cause little global signal in $R_{NET}$ (Byrne and Schneider, 2018). Given the robust performance
in CMIP models and generally analogous performance between the CCF configurations, we reaffirm that this
analysis framework is applicable for modelled radiative anomalies as well as observed.

### 5.3 Physical interpretation of the cloud-radiative sensitivities

In addition to the statistical performance metrics shown above, we study the spatial distribution and
magnitude of the sensitivities. Interpreting spatial sensitivities can be used in CCF analysis to justify predictor
selection that is grounded in physical reasoning and can be done for any of the CCF configurations (e.g., Andersen
et al., 2023). In this section, we physically interpret the sensitivities of $R_{LW}$ to the CCFs in configuration $S_{UT} +$
$\Delta U_{300}$ using a 21x11 domain, derived from observations and CMIP models (shown in Fig. 7). For local $R_{LW}$
predictions, many of the configurations perform similarly. However, sensitivities derived in this type of
framework are widely used to constrain global feedback, which leads us to put more emphasis on our results for
globally-aggregated $R_{LW}$ predictions (Fig. 2 and 5, c-d). Here, we found more marked differences between the
configurations, with $S_{UT} + \Delta U_{300}$ (at the optimal 21x11 domain size) superseding other configurations for $R_{LW}$
and performing well for $R_{NET}$ (only behind configuration $S_{UT}$). Though we only show spatial sensitivities for
configuration $S_{UT} + \Delta U_{300}$ here, for alternative configurations to be used in CCF applications, such as an
observational constraint on cloud feedback, we recommend similar physical interpretation of their sensitivities be
performed.

For each CCF in the configuration, we sum each contribution $\theta_i$ within the entire spatial domain (e.g.,
Eq. (5) for $R_{LW}$) and plot the total for each grid-cell. This is the spatial sensitivity of the cloud-radiative anomaly
to a given CCF, normalised for a one-standard deviation anomaly. Here, we derive the sensitivities using the full
twenty-year datasets (with no dataset rotation). There are several studies interpreting relationships between cloud-
radiative anomalies and the core CCFs (e.g., CN21, Andersen et al., 2023), though not explicitly for high clouds.



Therefore, we first briefly interpret our sensitivities to the core CCFs, shown in Fig. 7a-d. We then assess the sensitivities for cloud properties (i.e., cloud top pressure and cloud fraction) before interpreting sensitivities for the additional CCFs, $S_{UT} + \Delta U_{300}$.

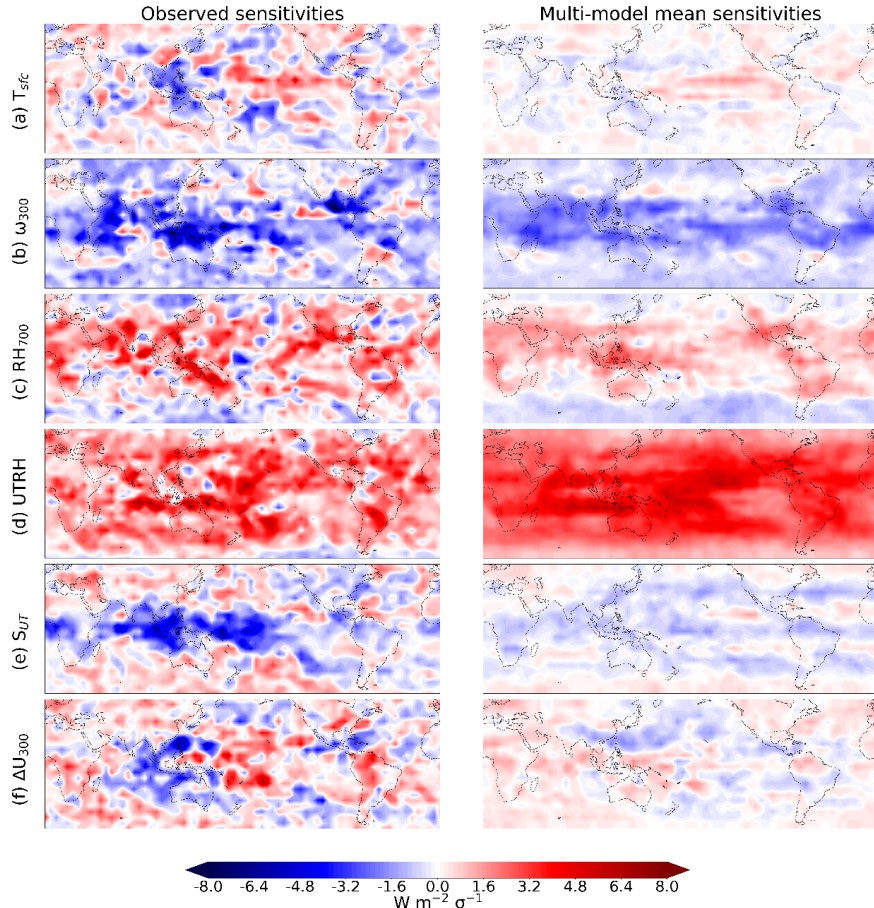

**Figure 7. $R_{LW}$ sensitivities ($\sum \theta_i$) to the cloud controlling factors in configuration $S_{UT} + \Delta U_{300}$ (also with $T_{sfc}$, $RH_{700}$, UTRH and $\omega_{300}$), derived using a 21x11 domain and defined for a one-standard deviation anomaly in each CCF (scaled using ERA5 CCFs for visualisation purposes). To produce the maps, we sum all elements of the sensitivity vectors at each point $r$. The left column shows observed sensitivities, and the right shows the multi-model mean.**

The observed and multi-model mean spatial distributions for the core CCFs – $T_{sfc}$, $\omega_{300}$, UTRH and $RH_{700}$ – broadly align what we expect, and are qualitatively similar between the observations and multi-model means. For all CCFs except UTRH, the magnitude of the modelled sensitivities are smaller than the observed (not shown globally, though tropical ascent means are shown in Fig. 8). We note that the observed global median regularization parameter, α, lies towards the upper-end of the inter-model spread, exceeded by only three models

(CanESM5, CanESM2, and MIROC-ESM; see Fig. S6). We speculate that the CCFs in the CMIP models typically capture the variability in $R_{LW}$ with greater skill than the observations, thus requiring less regularization. It is also known that (CMIP5) GCMs underestimate the frequency of tropical anvil cloud and extratropical cirrus occurrence (Tsushima et al., 2013; Ceppi et al., 2017), and thus their radiative effects. This may also be responsible





for smaller sensitivities on average. We also note that sensitivities do not explicitly imply causality, as certain
relationships are known to be bidirectional (e.g., $R_{LW}$ and UTRH).

The $R_{LW}$ - $T_{sfc}$ sensitivities (i.e., Eq. (5) summed for all $X = T_{sfc}$), shown in Fig. 7a, are generally small
in magnitude, with regions of positive sensitivity in the Central and East Pacific, and with negative (or, for the
CMIP models, a reduced magnitude) sensitivity over the Maritime continent. The $R_{LW}$ - UTRH sensitivities are
ubiquitously positive, and large in magnitude, consistent with increasing high-cloudiness with humidity, though
Fig. 7d suggests that CMIP-modelled sensitivities are larger in magnitude than is observed. We speculate that this
may be due to stronger coupling between upper-tropospheric humidity and cloud incidence in the CMIP models
than in the observations (perhaps owing to the parameterization of clouds in the models (Li et al., 2012; Qu et al.,
2014)). The $RH_{700}$ sensitivities are also widely positive (though negative at high latitudes), with smaller magnitude
than UTRH (as we would expect for high clouds) and the largest magnitudes in the deep tropics. Indicating
increased high-cloudiness with increased ascent, the $\omega_{300}$ sensitivities are near-ubiquitously negative, with the
strongest magnitudes broadly aligning with the tropical ascent regions in both observations and the CMIP models.

We can also use the decomposition of $R_{LW}$ into its linear sum of contributions from changes in cloud top
pressure ($CTP$), cloud fraction ($CF$), optical depth, and a small residual (with other components held fixed), to
further interpret our sensitivities (Zelinka et al., 2012a, b, 2016). We do not show optical depth sensitivities here,
owing to their small role in driving LW high-cloud radiative anomalies (see Fig. S9). LW radiative anomalies
caused by changes in the cloud properties are henceforth referred to using an additional subscript, i.e., $R_{LW,CTP}$ is
the contribution that changes in cloud top pressure (with no change in $\tau$ or $CF$) have on the total $R_{LW}$. Sensitivities
for the decompositions can be found in the Fig. S7. We average the domain-summed sensitivities in the tropical
ascent regions, shown in Fig. 8.






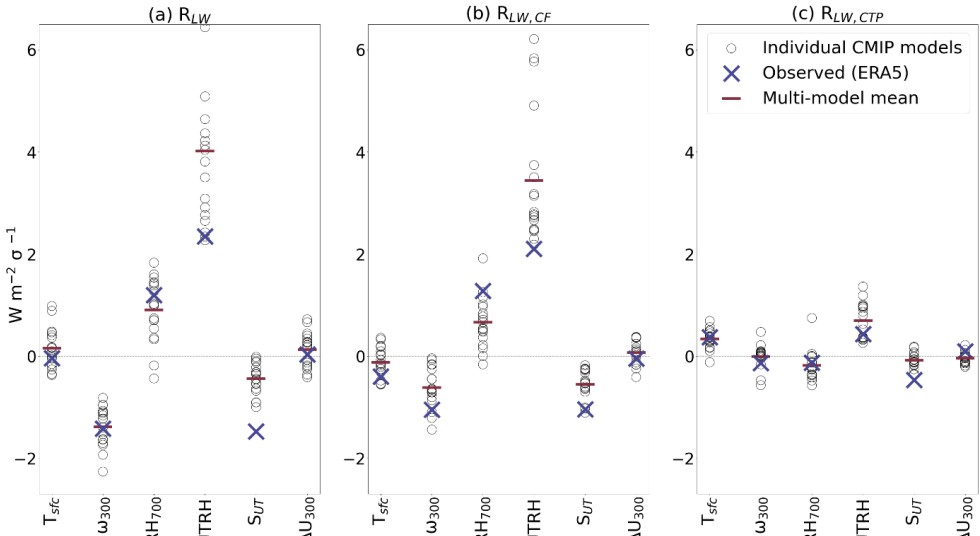

**Figure 8.** Observed and CMIP sensitivities to the cloud controlling factors in configuration $S_{UT} + \Delta U_{300}$ (with $T_{sfc}$, $RH_{700}$, UTRH and $\omega_{300}$), derived using a 21x11 domain and defined for a one-standard deviation anomaly for each CCF, averaged over all tropical ascent grid-cells for (a) $R_{LW}$, (b) $R_{LW,CF}$, and (c) $R_{LW,CTP}$. The standard deviation used to scale each CCF has been calculated from the observed CCFs.

The $R_{LW}$ - $T_{sfc}$ sensitivities average to approximately zero in the tropical ascent regions for both observations and CMIP models (Fig. 8a). However, Fig. 7a shows a distinct positive sensitivity present over the Pacific Ocean, which we ascribe to an increase in high-cloud top pressure that is associated with warming sea surface temperature anomalies, thus radiating heat to space at cooler temperatures. Indeed, we find that the spatial

pattern of $R_{LW,CTP}$ - $T_{sfc}$ sensitivities in this area are predominantly positive (Fig. 8c, Fig. S7), as we would expect. Accordingly, the mean $R_{LW,CTP}$ - $T_{sfc}$ sensitivity in the tropical ascent regions is positive, with larger magnitude than the similarly averaged and opposite-signed $R_{LW,CF}$ - $T_{sfc}$ sensitivities (Fig. 8). This is despite a much smaller monthly signal for $R_{LW,CTP}$ than $R_{LW,CF}$.

The $R_{LW}$ - $\Delta U_{300}$ sensitivity, shown in Fig. 7f, is more challenging to interpret than the core CCFs (Anber

et al., 2014). This is partially due to the dynamic nature of wind shear; coefficients within the spatial domain capture dynamic variability signals, which may result in a range of positive and negative sensitivities, therefore cancelling in the summation over the 21x11 domain. Nonetheless, we suggest reasons for both positive and negative sensitivity. There is also less agreement between the observed and multi-model mean spatial distributions than all other CCFs, which we speculate may partially be caused by offset circulation cells in the CMIP models,

resulting in different local sensitivities and dynamic signals. Over the Maritime Continent and Indian Ocean, observed sensitivities are broadly negative. It is known that wind shear can hasten the dissipation of tropical tropopause cirrus (Jensen et al., 2011) which would result in decreased cloudiness, and thus LW cooling. Conversely, there are many regions where the sensitivity is positive (such as the Central Pacific) which indicates LW warming with increased shear. We speculate that this is a result of shear spreading the high cloud, thus

increasing cloud fraction (Lin and Mapes, 2004), and in turn reducing outgoing LW radiation. The role of wind shear may be sensitive to the pressure level relative to the tropopause (Chakraborty et al., 2016; Nelson et al., 2022). Given that we use the same shear height (i.e., the difference in 300 hPa and 925 hPa wind speeds) globally,



we speculate the zonal distribution of tropopause heights may cause the differing relationships. Despite differences between the spatial distributions, both observed and the multi-model mean sensitivities in the tropical ascent regions are consistent with each other, and average to approximately zero (Fig. 8), which we might suspect given that shear is important for the organisation of convection which is not represented in GCMs.

Finally, we address the $S_{UT}$ sensitivities. Both the observed and multi-model mean $R_{LW}$ - $S_{UT}$ sensitivities, shown in Fig. 7e, are predominantly negative, with largest magnitudes over the Central and West Pacific, and Maritime Continent (though more markedly so for the observations). Therefore, in the absence of changes in the other CCFs, anomalies in high cloud associated with an increase in $S_{UT}$ would result in increased longwave emission to space over the tropics. This is what we expect, given the negative relationship between upper-tropospheric cloud incidence and static stability over tropical oceans (Li et al., 2014).

The $R_{LW,CF}$ - $S_{UT}$ sensitivities are also negative across the tropics (see Fig. S7), revealing that LW cooling arises – at least in part – from a reduction in high-cloud fraction, particularly in observations. This qualitatively resembles the anvil iris mechanism (Bony et al., 2016; Saint-Lu et al., 2020). As anvil clouds rise in response to global warming, their environment becomes more stable, owing to the dependency of static stability on atmospheric pressure (Saint-Lu et al., 2020, 2022). In a more stable atmosphere, the vertical pressure gradient associated with subsidence in clear-sky conditions is reduced. Owing to mass conservation, a reduction in the subsidence pressure gradient results in a reduction in anvil cloud fraction, caused by a decrease in horizontal convergence (Saint-Lu et al., 2020, 2022).

Though it is thought to be small in magnitude, the anvil cloud area feedback is subject to much uncertainty and considered to be underestimated by GCMs (Zelinka et al., 2022). Consistent with this, we find that the magnitude of the CMIP $R_{LW}$ - $S_{UT}$ sensitivities in the tropical ascent regions is substantially smaller than the observed (Fig. 8a). Indeed, the observed tropical ascent mean $R_{LW}$ - $S_{UT}$ sensitivity lies below the range of the individual CMIP-models' sensitivities. A large fraction of the total $R_{LW}$ - $S_{UT}$ sensitivity arises from the $CF$ component (Fig. 8b), which is consistently underestimated by the CMIP models. This is unsurprising; CMIP models have been  shown to underestimate the negative anvil cloud fraction feedback (Zelinka et al., 2022). Given that static stability has been shown to robustly control high-cloud fraction (Saint-Lu et al., 2022), and based on our results, we therefore speculate that the addition of $S_{UT}$ into observational constraint frameworks may reduce some of the uncertainty arising from the anvil fraction feedback.

We also find that the spatial distributions for the $R_{LW,CTP}$ - $S_{UT}$ observed and multi-model mean sensitivities are similar to each other, mostly negative, and largest in magnitude in the tropics. This suggests that an increase in $S_{UT}$ results in LW cooling, arising from a change (i.e., a decrease) in cloud top pressure. Increased static stability results in suppressed vertical motion, which in turn prevents cloud tops from rising as high as they might in a more unstable environment (Zelinka and Hartmann, 2010, 2011; Saint-Lu et al., 2022). Though the spatial distributions are similar, the magnitude of the CMIP $R_{LW,CTP}$ - $S_{UT}$ sensitivities are once again smaller than the observed in the tropical ascent regions (Fig. 8c).

As well as absorbing upwelling LW radiation, high clouds can reflect incident SW radiation depending on their optical depth. While the $R_{LW}$ (and thus the sensitivities) is primarily driven by $CF$ and $CTP$ changes, $R_{NET}$ is also driven by changes in optical depth, which predominantly affects SW radiative anomalies that we have not directly assessed. Thus, the net high-cloud radiative anomaly is comprised of complex interplay between competing LW and SW effects. However, we note that the magnitude of the observed $R_{NET}$ - $S_{UT}$ sensitivity is



smaller in magnitude than the $R_{LW} - S_{UT}$ component, though the spatial distribution is similar (Fig. S8). Therefore, assuming an increase in $S_{UT}$ with warming (Bony et al., 2016), the observed sensitivity suggests that

the stability iris mechanism is dominated by the LW response, and thus contributes a negative net feedback. In contrast, the magnitude of the multi-model mean $R_{NET}$ - $S_{UT}$ sensitivity is notably smaller in comparison to the other CCFs. The multi-model mean sensitivities therefore suggest little change in anvil cloud fraction with increasing $S_{UT}$, and therefore only a very small net cloud feedback. This again reflects the known underestimation of the negative anvil cloud fraction feedback in CMIP models (Zelinka et al., 2022). Additionally, Zelinka et al.

(2022) show that eight CMIP models (including six of those used in this research) predict an "unlikely" positive feedback arising from changes in anvil cloud fraction. Therefore, the near-zero multi-model sensitivities may also arise due to cancellation of local sensitivities between the models.



### 5.4 Predicting radiative anomalies from cloud fraction and cloud top pressure changes

Based on the physical interpretation of the sensitivities, our results – combined with previous literature and theory –
thus far support the use of $S_{UT} + \Delta U_{300}$ in high-cloud controlling factor frameworks. We have shown that $S_{UT} + \Delta U_{300}$
reproduces the globally-aggregated $R_{LW}$ time series with the highest skill in both observations and CMIP models, and the
sensitivities shown in Fig. 7 suggest that the mechanisms driving high-cloud feedback – rising free-tropospheric clouds and
reduction in anvil cloud fraction (Ceppi et al., 2017) – are captured by this selection of CCFs. Therefore, we finally question
whether our approach captures the spatial pattern, temporal variability, and magnitude of these properties.

We predict twenty years of cloud-radiative anomalies induced by $CF$ and $CTP$ changes (with other components held
fixed) for both observations and CMIP models, again using rotating eighteen-year datasets. The monthly radiative anomalies
are aggregated globally and in tropical ascent regions (e.g., as in Figs. 2 and 4, c-d) and compared against similarly aggregated
target values using the Pearson $r$ correlation coefficient (to ensure trends are captured by our framework). We do not assess
optical depth-induced changes, owing to their small historical LW signal (see Fig. S9). Though optical depth is important for
historical SW (and consequently, net) radiative anomalies, the high-cloud optical depth feedback is thought to be relatively
small (Zelinka et al., 2022) and so we focus on $CF$ and $CTP$. We place particular emphasis on the observations here, as the El
Niño phase of the El Niño-Southern Oscillation (ENSO) from July 2015 to June 2016 saw anomalous warming in the East
Pacific (see Fig. 9, top panel). ENSO is a dominant driver of natural ocean-atmosphere variability, resulting in regional tropical
temperature and circulation anomalies that are accompanied by changes in cloud properties and the TOA radiation budget
(Ceppi and Fueglistaler, 2021). Accordingly, July 2015 to June 2016 has one of the most anomalously warm annual mean
surface temperatures in the 20-year record. We only highlight this El Niño event for the observed cloud properties, as it will
be absent from the coupled historical simulations, and AMIP simulations do not reach 2016.



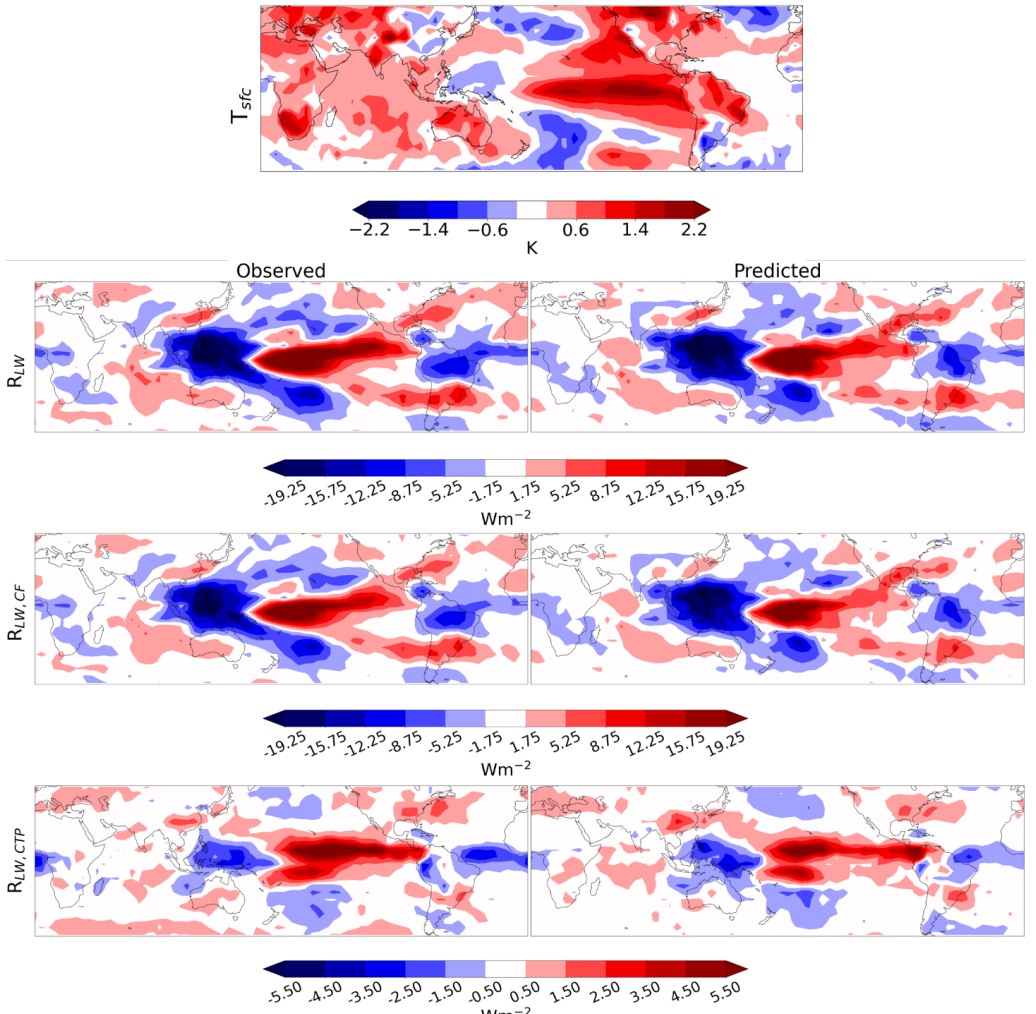

**Figure 9. Observed mean El Niño surface temperature anomaly (top) and radiative anomalies (left panels), averaged from July 2016 – June 2015 relative to the full twenty-year record. Predicted anomalies (right panels) made using a 21x11 domain and the configuration $S_{UT} + \Delta U_{300}$ (with T$_{sfc}$, RH$_{700}$, UTRH and ω$_{300}$) for the El Niño months.**

Out-of-sample globally-aggregated $R_{LW,CF}$ is predicted well using the $S_{UT} + \Delta U_{300}$ configuration (Fig. 10a). The

spatial distribution of the El Niño $CF$ anomalies (shown in Fig. 9) is also well reproduced, with a correlation coefficient of $r =$

0.93. There is a positive $R_{LW,CF}$ anomaly in the East Pacific, overlapping the region of anomalous sea surface warming,

indicating increased cloud fraction. Warmer SSTs enhance convection, resulting in increased upward motion, and thus

increased high cloudiness. In the West Pacific, the SST anomaly is negative and smaller in magnitude, though there is a strong,

negative $R_{LW,CF}$ anomaly, indicating a reduction in cloud fraction. Owing to the shift in circulation, suppressed convection can

result in anomalous subsidence, hence reducing high cloudiness.

We also predict observed, globally-aggregated $R_{LW,CTP}$ well, though with slightly reduced correlation coefficients

compared to $R_{LW,CF}$. Figure 10b shows that the magnitudes of strong positive and negative anomalies are slightly

underestimated; this may be caused by a small signal for the regression model to learn from (see Fig. S9). This may

alternatively hint towards a non-linear relationship between cloud top pressure and the CCFs, which would not be captured by





ridge regression. Regardless, the spatial distribution of predicted El Niño $R_{LW,CTP}$ is again strongly correlated to the observed, here with $r = 0.81$ (shown in Fig. 9). Strong positive anomalies are present over the East Pacific, which we ascribe to a rise in cloud top pressure due to enhanced convection. As the atmosphere warms, a shift of the $R_{LW,CTP}$ distribution towards higher values, particularly in the tropics, may be expected owing to the rising of free-tropospheric clouds (Ceppi et al., 2017). We note that the globally-aggregated, annual mean $R_{LW,CTP}$ during this El Niño event is most extreme, positive anomaly in the

observed twenty-year record, and is reproduced with small absolute error (0.003 Wm σ⁻²). This is consistent with the extreme warmth during that period, and the associated rise of the tropopause. Despite potentially underestimating the amplitude of the monthly variability, our method does an excellent job capturing the most extreme positive anomaly out-of-sample (globally-aggregated shown in Fig. 10; for tropical ascent aggregation, see Fig. S10).

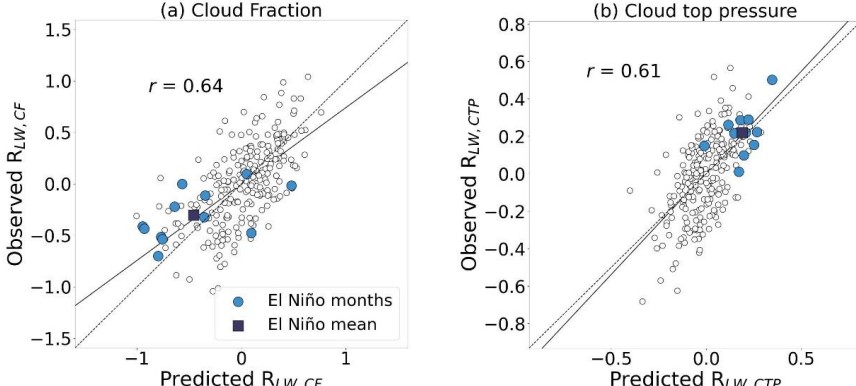

**Figure 10. Scatter plot showing the correlation between observed and predicted monthly globally-aggregated (a) $R_{LW,CF}$ and (b) $R_{LW,CTP}$ time series using configuration $S_{UT} + \Delta U_{300}$ (in addition to T$_{sfc}$, RH$_{700}$, UTRH, and ω$_{300}$) and a 21x11 domain. El Niño months are shown using coloured circles, with the annual mean shown using a coloured square. Solid lines show y = x, and the dashed lines show the line-of-best fit through the points. For results in the tropical ascent regions, see Fig. S10.**

We additionally confirm that our framework predicts out-of-sample globally-aggregated $R_{LW,CF}$ and $R_{LW,CTP}$ with good skill in the CMIP models, once again with slightly higher correlation coefficients than the observed (multi-model medians of 0.81 and 0.78, respectively; see Fig. S11). Though we only show results for the $S_{UT} + \Delta U_{300}$ configuration here, we have additionally assessed each configuration's ability to capture both $R_{LW,CF}$ and $R_{LW,CTP}$ for each month during the 2015 – 2016 El Niño (see coloured circles in Fig. 10 scatterplots). In a comparison of the El Niño months alone, we find that any

configuration including $S_{UT}$ reproduces tropically averaged $R_{LW,CTP}$ and $R_{LW,CF}$ anomalies with stronger positive correlations than alternative CCFs (not shown). We reiterate that the $S_{UT}$ sensitivities (Fig. 7e, Fig. S7) are physically congruent with previous literature, and appear to directly target the drivers of high-cloud feedback. We have additionally shown that the spatial distribution of the observed El Niño anomalies are captured well, including the extreme positive $R_{LW,CTP}$ anomaly, thus highlighting the strength of $S_{UT}$, in particular, as a high-cloud controlling factor.

**6 Conclusion**

A selection of candidate cloud controlling factors (CCFs) has been used to predict high-cloud radiative anomalies using ridge regression. We investigate five candidate CCFs: static stability in the upper troposphere, sub-cloud moist static energy, wind shear, convective available potential energy and convective inhibition, using the additional "core" meteorological drivers surface temperature, lower- and upper-tropospheric relative humidity, and upper-tropospheric vertical pressure velocity

in each configuration. CCFs are used within a two-dimensional spatial domain to predict out-of-sample longwave cloud-radiative anomalies, $R_{LW}$. We assess configurations from local to globally-aggregated spatial scales, and physically interpret



the spatial distribution of the sensitivities for the configuration $S_{UT} + \Delta U_{300}$. Finally, we assess the skill of $S_{UT} + \Delta U_{300}$ for predicting out-of-sample anomalies induced by changes in cloud top pressure and cloud fraction, including the El Niño event of 2015 – 2016.

We find that the optimal domain size and CCF combination is dependent on the temporal and spatial scales assessed, and we summarise the most relevant findings here:

1.  All configurations predict out-of-sample historical variability for both $R_{LW}$ (and $R_{NET}$) anomalies with good skill for observations and CMIP models at local scales. A domain of 7x3 optimises local predictions, where we show that ridge regression skill surpasses traditional multiple linear regression.

2.  Converse to local predictions, predictive skill for globally-aggregated radiative anomalies *increases* with domain size, peaking at 21x11. Unravelling this domain size discrepancy between local and global predictions remains a key question for future research. Differences between the configurations are more pronounced at global scales, and based on these results we identify $S_{UT} + \Delta U_{300}$ as an optimal configuration that performs well at all scales.

3.  The main mechanisms driving high-cloud feedback – rising of free-tropospheric clouds and reduction of anvil cloud
fraction – appear to be captured by the sensitivities in the $S_{UT} + \Delta U_{300}$ configuration. The spatial distributions of the $R_{LW}$ sensitivities to the core CCFs and $S_{UT}$ are physically consistent with our understanding and expectations, with observed and CMIP-modelled sensitivities qualitatively similar. There are larger differences between observed and the multi-model mean $\Delta U_{300}$ sensitivities, which are more complex to interpret than the core CCFs and $S_{UT}$.

4.  Out-of-sample globally-aggregated anomalies induced by cloud top pressure and cloud fraction changes are predicted
well using $S_{UT} + \Delta U_{300}$, in both observations and models. In particular, we obtain a quantitatively accurate out-of-sample prediction of the observed extreme anomalies in $R_{LW}$, $R_{LW,CF}$ and $R_{LW,CTP}$ during the 2015 – 2016 El Niño. The corresponding spatial distributions are also predicted with high correlation coefficients ($r > 0.80$).

Our systematic evaluation of high-cloud controlling factors highlights $S_{UT}$ as an important addition to CCF frameworks. Of course, our work is only the first attempt to assess candidates for high-cloud controlling factors so we welcome
future work on additional candidate factors that might not have been considered here. We have also identified an important inconsistency regarding ideal domain size for CCF predictions on historical data locally, and globally aggregated. Given the strong out-of-sample predictive power of our framework, in future work we will use our optimal CCF configurations to constrain high-cloud feedback.

**Data availability**

ERA5 meteorological reanalysis data is freely available from the Copernicus Climate Change Service (C3S) Climate Data Store (Hersbach, H., *et al*., 2023a, DOI: 10.24381/cds.f17050d7, Hersbach, H., *et al*., 2023b, DOI: 10.24381/cds.6860a573 and Hersbach, H., *et al*., 2023c, DOI: 10.24381/cds.bd0915c6). Combined MODIS Aqua/Terra data are also freely available and downloaded monthly (Bodas-Salcedo *et al.*, 2011, DOI: 10.5067/MODIS/MCD06COSP_M3_MODIS.062). All CMIP5/6 data were obtained from the UK Center for Environmental
Data Analysis portal (https://esgf-index1.ceda.ac.uk/search/cmip6-ceda/).

**Acknowledgements**

S.W.K., P.N., P.C. and P.S. were supported through the UK Natural Research Environment Research Council (NERC) grant number NE/V012045/1. H.A., J.C. and P.S. have received funding from the European Union's Horizon 2020 research and innovation program under grant agreement no. 821205 (FORCeS) and H.A. and J.C. from the Deutsche
Forschungsgemeinschaft (DFG) in the project Constraining Aerosol–Low cloud InteractionS with multi-target MAchine



learning (CALISMA) under project number 440521482. This research was carried out on the High Performance Computing Cluster supported by the Research and Specialist Computing Support service at the University of East Anglia and additionally JASMIN, the UK's collaborative data analysis environment (https://jasmin.ac.uk).


**Author contributions**. S.W.K., P.N., and P.C., conceptualised this research. S.W.K. conducted the analysis and wrote the article with contributions from all authors on the text and interpretation of the results.

**Competing interests**. At least one of the (co-)authors is a member of the editorial board of Atmospheric Chemistry and Physics.

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
