# Peer review of "A systematic evaluation of high-cloud controlling factors"

_EGUsphere, 2024_

## Author Comment (AC1)

We are grateful for the reviewer's perceptive and constructive feedback, which has substantially enhanced this manuscript. We are confident that this research has been strengthened following their beneficial comments. Our responses to their principal points are outlined below:

*"In this manuscript, the authors use ridge regression to extend the work of CN21 to analyse LW cloud radiative anomalies in the form of Cloud Controlling Factor (CCF), i.e. relationships between large-scale satellite cloud observations and meteorological predictors. Logically, they focus on high clouds. They define a set of four "core CCFs" and of about ten "candidate CCFs". The authors present the method, variables and data in the first four sections, followed by the results in the fifth. I'm not expert enough in their ridge regression method to give a relevant opinion on the statistical validity of their work, and I consider it relevant.*

*The skills metrics for the various indicators are examined in sections 5.1 and 5.2, in particular on the basis of Figures 2 and 5. I find this part of the manuscript particularly problematic. The most systematic and robust variation of these CCFs is as a function of the size of the domains considered, and this aspect is little discussed"*

**C1.** There is indeed robust variation of predictive skill with domain size, we have since expanded our discussion into the evolution of skill with domain, though we are keen to keep the focus on differences between the CCFs. To address this comment, we have included a new figure in the supplementary (Fig. S6) and expanded Section 5.1.1 with the following additions:

- Composite sensitivities have been analysed (with 2 shown in the supplementary Fig. S6 and shown below, and Figure R1). Composite sensitivities are constructed by domain-averaging independent local 21x11 sensitivities, where the target grid-cell matches some criteria (e.g., averaging all local sensitivities for West Pacific tropical ascent grid-cells).
- Explained value fraction (EVF) has been assessed at 3 different domain sizes (also included in Fig. S6).

Composite sensitivities (**see R1 and Fig. S6 shown below**) have been visually inspected, where we find that there appears to be a distinction between more "local" (such as UTRH) and more "non-local" (such as $T_{sfc}$) predictors. We find that the local predictors have largest magnitude concentrated close to the target grid-cell and a decreasing EVF with domain size. Conversely, "non-local" predictors have increasing EVF and more spatially coherent patterns beyond the target grid-cell. For local predictors, the sensitivities appear noisier away from the target grid-cell – there appears to be much "salt and pepper" style noise.

[Figure]

**Figure S6. Composite spatial sensitivities using the 21x11 domain and configuration $S_{UT} + \Delta U_{300}$ (with additional core CCFs $T_{sfc}$, RH₇₀₀, UTRH, and ω₃₀₀) in (a) tropical ascent grid-cells (defined by climatological mean EIS < 1 K, and ω₅₀₀ < 0 hPa s⁻¹) in the East Pacific (130°W to 80°W) and (b) North Atlantic (60°W to 10°E, latitudes north of 30°N) midlatitude clouds (climatological mean EIS > 1 K, and ω₅₀₀ < 1.5x10⁻⁴ hPa s⁻¹). Panel (c) shows the global mean EVF as a function of cloud controlling factor and domain size for local predictions. Note that the global mean EVF has only been weighted based on latitude, and not as a function of $R_{LW}$ standard deviation. Panel (d) shows the EVF for globally-aggregated predictions.**

[Figure]

*R1. Composite spatial sensitivities smoothed using a Gaussian filter ($\sigma = 2$) to reduce non-local noise. The 21x11 domain and configuration $\boldsymbol{S_{UT}} + \Delta\boldsymbol{U_{300}}$ has been used (with additional core CCFs $\boldsymbol{T_{sfc}}$, $RH_{700}$, UTRH, and $\omega_{300}$) for (a) tropical ascent grid-cells (defined by climatological mean EIS < 1 K, and $\omega_{500} < 0$ hPa s$^{-1}$) in the East Pacific (130°W to 80°W) and (b) North Atlantic (60°W to 10°E, latitudes north of 30°N) midlatitude clouds (climatological mean EIS > 1 K, and $\omega_{500} < 1.5x10^{-4}$ hPa s$^{-1}$). These regions are the same shown in Figure S6.*

We propose that an increasing domain size can incorporate more relevant non-local contributions for CCFs such as $T_{sfc}$, $S_{UT}$, and $\Delta U_{300}$, whilst also adding potentially less relevant information for more localised CCFs such as UTRH, $\omega_{300}$ and $RH_{700}$ (which we find contribute a large proportion of the local EVF). Thus, predictive skill is a trade-off between additional relevant non-local context (e.g., from $T_{sfc}$) and the inclusion of less-relevant information (e.g., from UTRH), increasing the domain size and thus the dimensions of the model. We have referenced the findings of this analysis in the bullet point list starting at line 417:

"

- *There is an emergent distinction between "local" and "non-local" predictors. For example, EVF for UTRH decreases with increasing domain size and, accordingly, we find that local UTRH sensitivities typically have strong magnitudes close to target grid-cell, with noisy, spatially incoherent coefficients further afield (see Fig. S6a-b for an example); thus, we describe UTRH as a "local" CCF (similarly for $\omega_{300}$ and $RH_{700}$).*

- *EVF for $T_{sfc}$, $\Delta U_{300}$, and $S_{UT}$ increases with domain size (i.e., "non-local" predictors), and each contribute a greater proportion of the globally-aggregated predictions compared to local predictions (Fig. S6c-d).*

- *Predictive skill is likely a trade-off between adding relevant information from "non-local" CCFs while adding superfluous information from "local" CCFs; i.e., too distant information does not provide additional predictive skill, at least to the degree that it would outweigh the corresponding increase in dimensionality of the regression problem.*

- *For globally-aggregated predictions, $\omega_{300}$ is the least important predictor (compared to the second most important for local predictions), thus explaining why the choice of pressure level of $\omega$ is less relevant at global scales (shown in Fig. 4) than locally.*

*"*

We have also included a new section in the supplementary material that discusses the discrepancies in domain size in more detail (Section S3).

*"Most of the discussion focuses on the variation of the metrics according to the CCF. But, for the same size of domain, these variations are extremely small, of the order of 1% for the RMSE of local predictions, of 10% for the Pearson number of the integrated value of predictions, i.e. often of the order of the last digit given. **There is no discussion as to whether this level of precision is relevant.** Some variations may be different depending on the size of the domain. "*

**C2.** We agree with the reviewer's comment; the changes in skill between the configurations indeed might on first sight appear small (although it is important to highlight again that predictive skill becomes increasingly hard to gain as the baseline performance is increased towards perfect scores), and thus determining the statistical relevance of these changes is important. Therefore, in addition to showing the metrics for the single 20-year observed time series (which we showed in main text Fig. 2) we have since statistically tested the differences in predictive skill between the configurations at the optimal domains for local (7x3) and globally-aggregated (21x11) predictions using the Kruskal-Wallis test. To ensure statistical testing between configurations is robust, we construct a distribution of predictive skill using 100 bootstrapped samples. All references to statistical testing in this response (C2) consider observations only (C3 looks at the CMIP models).

[Figure]

*Figure R2. Schematic showing the process followed for constructing distributions of predictive skill.*

We separately train the ridge regression model using 216 months (equivalent to 18 years) of bootstrapped data (with replacement) and validate against any remaining months of the 20-year dataset. Training the model using only 216 months out of the total 240 ensures there are always at *least* 24 months of unseen data for validation (though typically this is around 80 due to repeated months in the training dataset), thus providing separate test and training datasets. This is repeated n = 100 times for each configuration. The 216 bootstrapped samples used to train the model are determined using the same random seed for each n-th iteration so that the same training datasets are used for comparison between each CCF configuration. The Pearson r is hence determined 100 times for each configuration from each test dataset with predicted values. This results in a distribution of the predictive skill for each configuration. A schematic of this process is shown above (R2).

We then perform a Kruskal-Wallis test for the 10 different configurations, locally and globally, to determine whether the distributions are statistically similar.

[Figure]

*R3. LHS: Boxplots showing the distribution of Pearson r values from 100 bootstrapped samples (where samples are a random 216 months) and validated against the remaining unseen months. Horizontal dashed line shows the mean value for configuration EIS (w300). RHS shows the PDF for each distribution, fit using kernel density estimation.*

Shown above is the distribution for *all* configurations (R3). We have added a new Figure to the main text (Figure 3) that shows a selection of configurations, demonstrating the evolution of predictive skill with CCF at a single domain dimension (see below). We first look at results for the observed cloud-radiative anomalies. For **local** predictions with 7x3 predictor dimensionality, there is a clear shift in the distributions changing $\omega_{500}$ to $\omega_{300}$. An additional shift occurs through the inclusion of shear. For local predictions, the Kruskal-Wallis test shows that there is sufficient evidence to suggest that the distributions of

predictive skill are different ($p < 10^{-30}$). Following this, we perform pairwise Kruskal-Wallis tests against the configuration with the highest median ($S_{UT} + \Delta U_{300}$). $S_{UT} + \Delta U_{300}$ is **only** statistically similar to configurations $m + \Delta U_{300}$ and $\Delta U_{300}$ at the 0.5 % ($p = 0.005$) significance level (smaller significance level to account for multiple statistical testing, following the Bonferroni correction: with $p = 0.06$ and $p = 0.01$, respectively). This has been included in Sect 5.1:

> *"To quantify whether differences between configurations are statistically significant for the observed anomalies, we generate a distribution of Pearson $r$ values using bootstrapping (Davison and Hinkley, 1997). We randomly sample the observed data (with replacement) 100 times, creating datasets equivalent in length to 18 years. Any remaining months are used as a validation dataset, where $r$ is determined against predicted values. This process results in a distribution of 100 $r$ values for each configuration, providing an estimate of predictive skill uncertainty, with a selection of the configurations shown in Fig. 3. The non-parametric Kruskal-Wallis test is hence used to identify statistical differences between all of the distributions. **We find highly significant differences between all of the configurations ($p < 10^{-30}$).** Accounting for its highest global median $r$, we pairwise test the predictive skill distribution for $S_{UT} + \Delta U_{300}$ with all other configurations (using an adjusted significance level of 0.5 % to account for multiple hypothesis testing). **We find statistical similarity with only $m + \Delta U_{300}$ and $\Delta U_{300}$ ($p = 0.06$ and $p = 0.01$, respectively).**"*

And for globally-aggregated anomalies, also in Sect. 5.1:

*"Here we find more marked improvements in predictive skill for most of the CCF configurations compared to EIS ($\omega_{500}$), with performance again strongly dependent on domain size (Fig. 2c-d). However, we now find that changing the pressure level of $\omega$ no longer results in a substantial positive shift of the skill distributions, though inclusion of $\Delta U_{300}$ still results in improvements (Fig. 3b). We also note that performance metrics for globally-aggregated $R_{LW}$ are comparatively worse than the globally-averaged local metrics. This is in line with accumulation of local errors and reduced variability in the globally-aggregated anomalies. In a comparison of all globally-aggregated distributions shown Fig. 3b,* **there is evidence showing statistical differences at the 5 % significance level (with $p < 10^{-40}$)**. *Here, $m + \Delta U_{300}$ has the highest median $r$. In a* **pairwise comparison of $m + \Delta U_{300}$ with each other predictive distribution, we find statistical differences with all configurations except $S_{UT} + \Delta U_{300}$ ($p = 0.02$) and $\Delta U_{300}$ ($p = 0.03$),** *here using the 0.5 % significance level owing to multiple statistical tests."*

[Figure]

**Figure 3. Box and whisker plots (left panels) showing the distribution of Pearson $r$ values based on 100 bootstrapped samples of $R_{LW}$ for a selection of the CCF configurations. Crosses show the means, notches show the medians, and circles show the outliers. A "CCF configuration" refers to the selection of cloud controlling factors used to predict $R_{LW}$, where each configuration uses $T_{sfc}$, $RH_{700}$, UTRH and $\omega_{300}$ (with the exception of the first box and whisker, where $\omega_{500}$ is used instead) and a candidate CCF(s) (e.g., $S_{UT}$), which is used to label each configuration. The right panels show the shapes of the distributions using kernel density estimation. The top panels (a) show the distributions for local predictions at the 7x3 optimal domain size (analogous to Fig. 2a) and the bottom panels (b) show the distributions for the 21x11 optimal domain size (analogous to Fig. 2c). EIS ($\omega_{300}$) is highlighted in black to facilitate easier comparison between configurations.**

*"There is a lot of discussion about how these indicators evolve according to the CCFs, but this evolution between CCFs is difficult to see.* **Given that it is mainly the dependence on CCFs that is being discussed, why not plot the values for a single domain dimension (as in Figure 6)?"**

[Figure]

**Figure 6. Pearson $r$ scores for (a) globally- and (b) tropical ascent-aggregated predictions made at the 21x11 domain size using different "CCF configurations". A "CCF configuration" refers to the selection of cloud controlling factors used to predict R$_{LW}$. Each configuration uses $T_{sfc}$, $RH_{700}$, UTRH and $\omega_{300}$ (with the exception of the first column, where $\omega_{500}$ is used instead) and a candidate CCF(s) (e.g., S$_{UT}$). The multi-model median Pearson $r$ is shown from the 14 CMIP models where CAPE and CIN is calculated. The bootstrapped ($n = 100$) predictive skill distributions for EIS ($\omega_{300}$) and $S_{UT} + \Delta U_{300}$ are shown at the optimal 21x11 domain size for (c) globally-aggregated predictions and (d) tropical-ascent aggregated predictions.**

**C3.** Thank you for your comment. To facilitate the ease of viewing evolution of configuration skill with CCFs, we have since included Figure 3 which shows the bootstrapped distributions of predictive skill for the "optimal" domain dimensions (7x3 and 21x11), making it easier to see the evolution of predictive skill for a single domain dimension. We have also removed the CMIP correlation matrix from the main text and included this in the supplementary instead (Fig. S7), while adding 2 additional panels to Figure 6 to show the evolution between EIS ($\omega_{300}$) and $S_{UT} + \Delta U_{300}$ individually for each CMIP model. We hope this better conveys the evolution of predictive skill between CCF configurations and also that predictive skill can be quite dependent on GCM. Given that the evolution of predictive skill with CCF configuration is quite dependent on GCM, we feel that the original heatmap (now Fig. S7) was not able to convey this complex information, and we believe our new Figure 6 better conveys that although median CMIP skill has very small variations, these variations are more pronounced at an induvial CMIP model level.

The discussion is included in Sect. 5.2:

*"In Section 5.1, we highlighted $S_{UT} + \Delta U_{300}$ as a possible optimal configuration. Here we identify whether differences between the CMIP-modelled predictive skill distributions for EIS ($\omega_{300}$) and $S_{UT} + \Delta U_{300}$ are statistically significant. In a pairwise Kruskal-Wallis test on the combined Pearson $r$ scores from all 18 models ($n = 1800$), we find a significantly higher predictive skill distribution for $S_{UT} + \Delta U_{300}$ than EIS ($\omega_{300}$) with $p < 10^{-11}$ (distributions not shown). This is unsurprising; 15 of the 18 individual CMIP models have a higher median $r$ using $S_{UT} + \Delta U_{300}$ compared with EIS ($\omega_{300}$).*

*Despite a slightly lower multi-model median, we find that the globally-aggregated distributions for all models combined are statistically similar at the 5 % significance level (shown in Fig. 6c, $p = 0.13$). Here, only half of the CMIP models have a higher median $r$ using $S_{UT} + \Delta U_{300}$ compared with EIS ($\omega_{300}$). However, visual inspection of the distributions for tropical ascent-aggregated predictions (Fig. 6d) suggests that improvements found using $S_{UT} + \Delta U_{300}$ instead of EIS ($\omega_{300}$) are more pronounced than any deteriorations. In summary, while the mean evolution of predictive skill within the CMIP models is broadly aligned with the observations, there are nuances which likely depend on the parameterization within the models themselves (Li et al., 2012; Qu et al., 2014; (Rio et al., 2019). This leads to a slightly different evolution of predictive skill with configuration between the CMIP models"*

*"These variations in performance according to the CCFs are generally very small, which leads the authors to make many suppositions but few assertions. The word "speculate" appears 15 times in the manuscript. Moreover, these small variations in performance according to the CCFs make many comments questionable in my opinion."*

We have expunged several instances of "speculate" and supported our suppositions with additional analysis. This additional analysis includes:

- The statistical testing of the differences between the configurations (see comments **C2 & C3)**
  - We show statistical differences between candidate CCF configurations and confirm differences are robust.

- Greater exploration of the domains-size discrepancy (see comment **C1**);
  - Increased discussion surrounding performance with domain size; we propose a trade-off in predictive skill exists between local and non-local CCFs (owing to increasing dimensions in the model).
- Improved visualisation of the CMIP sensitivities (see comment **C4**) and inclusion of correlation coefficients.

*"I have the same criticism of figure 7 and the associated comments. The observations are very noisy, as you would expect, which makes it difficult to compare the figures directly. **It would therefore be necessary to highlight what is significant and what is not, to show zonal averages, smoother results, and so on.**"*

**C4.** We again thank the reviewer for this helpful comment. We have added zonal average sensitivities alongside the spatial distributions, previously shown in Figure 7. We have added a similar panel to the Supplementary Figures S8 and S9. We have also included a correlation between the CMIP multi-model mean zonal sensitivity and the observed zonal sensitivity to more objectively discern whether the sensitivities are similar.

[Figure]

**Figure 7.** $R_{LW}$ sensitivities ($\sum \Theta_i$) to the cloud controlling factors in configuration $S_{UT} + \Delta U_{300}$ (also with $T_{sfc}$, $RH_{700}$, UTRH and $\omega_{300}$), derived using a 21x11 domain and defined for a one-standard deviation anomaly in each CCF (scaled using ERA5 CCFs for visualisation purposes). To produce the maps, we sum all elements of the sensitivity vectors at each point $r$. The column (a) shows observed sensitivities, and column (b) shows the multi-model mean. Column (c) shows zonal average sensitivity for the observations (dashed line), the multi-model mean (dark solid line) and individual CMIP model sensitivities. The Pearson $r$ correlation coefficient for the observed and CMIP multi-model mean zonal sensitivities is shown in the bottom corner of each panel.

*"In summary, this manuscript deals with an important subject and uses an original method of analysis, but needs major revisions in order to make the text less descriptive, reach more conclusions and ensure that these are better supported"*

We once again thank the reviewer for their thought provoking and helpful comments. We believe that the manuscript is much improved following their insightful questions and remarks.

**References**

Davison, A. C. and Hinkley, D. V.: Bootstrap Methods and Their Application, Cambridge University Press, 606 pp., 1997.

Li, J.-L. F., Waliser, D. E., Chen, W.-T., Guan, B., Kubar, T., Stephens, G., Ma, H.-Y., Deng, M., Donner, L., Seman, C., and Horowitz, L.: An observationally based evaluation of cloud ice water in CMIP3 and CMIP5 GCMs and contemporary reanalyses using contemporary satellite data, Journal of Geophysical Research: Atmospheres, 117, https://doi.org/10.1029/2012JD017640, 2012.

Qu, X., Hall, A., Klein, S. A., and Caldwell, P. M.: On the spread of changes in marine low cloud cover in climate model simulations of the 21st century, Clim Dyn, 42, 2603–2626, https://doi.org/10.1007/s00382-013-1945-z, 2014.

Rio, C., Del Genio, A. D., and Hourdin, F.: Ongoing Breakthroughs in Convective Parameterization, Curr Clim Change Rep, 5, 95–111, https://doi.org/10.1007/s40641-019-00127-w, 2019.

---

## Author Comment (AC2)

Reviewer 1

We appreciate the reviewer's insightful and constructive feedback, which has greatly enhanced the quality of the manuscript. We believe that the document has been significantly strengthened as a result of their thought-provoking questions and valuable remarks. We have responded to their key points as follows:

*"In this paper, ridge regression is used to systematically evaluate the addition of five candidate cloud controlling factors (CCFs) to previously established core CCFs within large spatial domains to predict longwave high-cloud radiative anomalies. The results show that upper-tropospheric static stability is an important CCF for high clouds and longwave cloud feedback. All combinations of tested CCFs perform quite well for most locations at grid-cell scales, while differences between configurations for predicting globally-aggregated radiative anomalies are more pronounced. The authors found that spatial domain size is more important than the selection of CCFs for predicting local anomalies, and there is discrepancy between optimal domain sizes for local and globally-aggregated radiative anomalies.*

*There are abundant technique details in the paper, and the method is potentially useful to evaluate the long-term high-cloud feedback. The paper might be accepted after addressing the following comments:*

*Specific comments:*

*1. In machine learning, **the dataset used to test the performance of a machine-learning model should be independent from the dataset that is used to train the data. What is the training dataset and testing dataset for the metrics of Fig. 2**? Ideally, the PI-control or AMIP simulations might be used as the training dataset, and abrupt4xco2 simulations might be used as the testing dataset.*

**C1.** We would like to reassure the reviewer that independent training and test datasets have been used throughout our analysis. We have used a rotating 2-year/18-year test/training dataset method to construct 20-years of predictions to compare against the full 20-year observed record. We have described our process with more clarity in the main text (line 296). We have also added a schematic (see Figure R1 below; this has been added to the supplementary, Fig. S2) that demonstrates our test/training process. This method is used consistently to determine the predictive skill shown in Figures 2 and S7 for both the observed and modelled data.

*"For Sect. 5.1, 5.2 and 5.4 we use sensitivities to predict a two years validation dataset. We repeat this process, rotating the withheld data every two years resulting in ten unique training-validation dataset combinations (**see Supplementary Fig. S2 for a schematic of this process).***

We use historical observations and historical simulations to assess selections of cloud controlling factors instead of AMIP or Pi-control simulations. This is because sensitivities derived from historical, observed data have been used to constrain GCM simulated abrupt-4xCO2 cloud feedback (e.g., Ceppi and Nowack, 2021; Myers et al., 2021, etc.,  see main text for others). It is thus important that the sensitivities themselves are validated on the historical

time series. For the most analogous comparison between observations and atmosphere-ocean coupled models, we therefore also use historical CMIP simulations.

[Figure]

*R1. Schematic demonstrating the process of rotating test-training datasets. This will be included in the supplementary material.*

> *"For observations or historical simulations, **the first several or more years might be used to train the ridge regression model, and the last several years might be used to test the performance of the model**"*

**C2.** We have chosen to use our rotating test-training dataset method to reduce the sensitivity of our skill metrics to outliers. For example, if we test on only the last 2 years, and a single extreme is underestimated, the $R^2$ score is affected strongly by the outlier (owing to a small sample size of 24 datapoints). Our validation procedure in general allows us to more robustly estimate and compare the relative performance of the various controlling factor selections.

> *"2. R-square and r are highly relevant metrics, so I suggest using only one of them in the main text."*

**C3.** We have expunged plots showing combined metrics so that Pearson $r$ is the only metric shown in the main text. $R^2$ and RMSE are briefly discussed in Section 5.1 and shown

in the supplementary (Fig. S4/S5). This is to show that low $R^2$ doesn't necessarily mean low Pearson $r$ or high RMSE.

*3. The **none-local effect of CCF on high cloud amount might be further explored and discussed**. The dependence of model performance to domain size might be associated with cloud transferring between adjacent grid boxes. In addition, previous studies suggest that the surface temperature in the tropics has significant impact on subtropical high cloud amount, is this process associated with the domain size dependence?*

**C4.** We have included further analysis and discussion regarding the non-local effect of CCF on high-cloud radiative anomalies. To summarise:

- Composite sensitivities have been analysed (with 2 shown in the supplementary Fig. S6 and shown below, and Figure R2). Composite sensitivities are constructed by domain-averaging independent local 21x11 sensitivities, where the target grid-cell matches some criteria (e.g., averaging all local sensitivities for Central Pacific tropical ascent grid-cells).
- Explained value fraction (EVF) has been assessed at 3 different domain sizes (also included in Fig. S6).

We have visually inspected composite sensitivities in conjunction with the EVF plots (**see R2 and Fig. S6 shown below**). Doing so, we find an emergent distinction between more "local" (such as UTRH, $\omega_{300}$ and $RH_{700}$) and more "non-local" (such as $T_{sfc}$, $\Delta U_{300}$ and $S_{UT}$) predictors. We find that our "local" predictors have decreasing EVF with domain size, whilst for non-local predictors the EVF increases with domain size. We also notice the non-local predictors have more spatially coherent patterns beyond the target grid-cell (e.g. wind shear in R2(b)), whereas for local predictors, the sensitivities appear noisier (much like "salt and pepper" noise) away from the target grid-cell, with largest magnitude concentrated close to the target grid-cell.

We propose that, while increasing domain size incorporates non-local contributions from $T_{sfc}$, $S_{UT}$, and $\Delta U_{300}$, the larger domain adds potentially less relevant information for the more localised UTRH, $\omega_{300}$ and $RH_{700}$ (also contributing a large proportion of the local EVF). The resultant predictive skill is thus a trade-off between additional relevant information (e.g., from $T_{sfc}$) and additional less-relevant information (e.g., from UTRH) which increases the domain size and thus the dimensions of the model. We have referenced the findings of this analysis in the bullet point list starting at line 412:

"

- *There is an emergent distinction between "local" and "non-local" predictors. For example, EVF for UTRH decreases with increasing domain size and, accordingly, we find that local UTRH sensitivities typically have strong magnitudes close to target grid-cell, with noisy, spatially incoherent coefficients further afield (see Fig. S6a-b for an example); thus, we describe UTRH as a "local" CCF (similarly for $\omega_{300}$ and $RH_{700}$).*

- *EVF for $T_{sfc}$, $\Delta U_{300}$, and $S_{UT}$ increases with domain size (i.e., "non-local" predictors), and each contribute a greater proportion of the globally-aggregated predictions compared to local predictions (Fig. S6c-d).*

- *Predictive skill is likely a trade-off between adding relevant information from "non-local" CCFs while adding superfluous information from "local" CCFs; i.e., too distant information does not provide additional predictive skill, at least to the degree that it would outweigh the corresponding increase in dimensionality of the regression problem.*

- *For globally-aggregated predictions, $\omega_{300}$ is the least important predictor (compared to the second most important for local predictions), thus explaining why the choice of pressure level of ω is less relevant at global scales (shown in Fig. 4) than locally.*
  *"*

We have also included a new section in the supplementary material that discusses the discrepancies in domain size in more detail (Section S3), and we also agree that there are several mechanisms that may cause non-local dependence on the CCFs. For example, increased static stability to the east of a target grid-cell may be advected locally, or indeed an adjacent cloud transferring grid-cells. This is included explicitly in Section S3.

*"Note that there are several mechanisms that may be associated with non-local sensitivities, including remote SST pattern effects for deep convection (Fueglistaler, 2019), the transferral of cloud from one grid-cell to another within the resolved time interval, or upstream/downstream advection of the meteorological drivers".*

*"Minor Comments:*

*It is recommended to check all instances of italicized text in the manuscript to ensure consistency throughout the text:*

*Line 99: delete the preposition "in".*

*Line 275: what is the variable "r"?*

*Line 287: where the second term on the right-hand side of Eq. (3) ...*

*Figure 1: The latitude and longitude coordinates should be marked on the map (and similarly for the subsequent figures).*

*Table 1: The formatting needs to be unified. For example, there are excessive gaps between certain words, and the sixth row of the table ("Key studies") lacks a space before it. Moreover, the last row ("Key studies") has a period, while the other rows do not."*

Thank you for pointing out the above corrections, which we have implemented.

*"Figure 9: I suggest adding an additional panel to compare the results with the CN21 method (i.e., CCFs containing only T_sfc, RH_700, UTRH, and ω_300). This comparison will better highlight the advantages of the new method."*

**C5.** Thank you for your suggestion. We have broadened the discussion on Figure 9 to more directly compare against alternative configurations. Though we have chosen to not include an additional panel in Figure 9 showing CN (as visually, the spatial distribution of the radiative anomalies are similar between EIS and $S_{UT} + \Delta U_{300}$), we have assessed the absolute prediction error for a range of configurations during El Niño (where we calculate error = predicted anomaly – observed anomaly, and total tropical absolute error is the absolute sum of all tropical error). We look at the *absolute* anomaly to avoid rewarding configurations that produce compensating positive and negative prediction errors which cancel when averaging. We have also included an additional panel (panel (c)) showing the spatial distribution of the anomalies.

We find that EIS ($\omega_{500}$ – the CN configuration) has the highest absolute prediction error for $R_{LW}$, $R_{LW,CF}$ and $R_{LW,CTP}$. In fact, including EIS actually *increases* the absolute anomaly relative to just the core CCFs. Conversely, configuration $S_{UT}$ (with no shear) has the lowest absolute error for $R_{LW,CF}$ and $R_{LW,CTP}$. For $R_{LW}$ and $R_{LW,CTP}$, $S_{UT}$ is followed by $S_{UT} + \Delta U_{300}$. These findings are mentioned in line 663:

*"We also find that configurations $S_{UT}$ and $S_{UT} + \Delta U_{300}$ predicts the tropical mean El Niño $R_{LW,CTP}$ with the smallest absolute error (not shown)"*.

[Figure]

**Figure S6. Composite spatial sensitivities using the 21x11 domain and configuration $S_{UT} + \Delta U_{300}$ (with additional core CCFs $T_{sfc}$, RH₇₀₀, UTRH, and ω₃₀₀) in (a) tropical ascent grid-cells (defined by climatological mean EIS < 1 K, and ω₅₀₀ < 0 hPa s⁻¹) in the East Pacific (130°W to 80°W) and (b) North Atlantic (60°W to 10°E, latitudes north of 30°N) midlatitude clouds (climatological mean EIS > 1 K, and ω₅₀₀ < 1.5x10⁻⁴ hPa s⁻¹). Panel (c) shows the global mean EVF as a function of cloud controlling factor and domain size for local predictions. Note that the global mean EVF has only been weighted based on latitude, and not as a function of $R_{LW}$ standard deviation. Panel (d) shows the EVF for globally-aggregated predictions.**

[Figure]

*R2. Composite spatial sensitivities smoothed using a Gaussian filter (σ = 2) to reduce non-local noise. The 21x11 domain and configuration $S_{UT} + \Delta U_{300}$ has been used (with additional core CCFs $T_{sfc}$, RH$_{700}$, UTRH, and $\omega_{300}$) for (a) tropical ascent grid-cells (defined by climatological mean EIS < 1 K, and $\omega_{500}$ < 0 hPa s$^{-1}$) in the East Pacific (130°W to 80°W) and (b) North Atlantic (60°W to 10°E, latitudes north of 30°N) midlatitude clouds (climatological mean EIS > 1 K, and $\omega_{500}$ < 1.5x10$^{-4}$ hPa s$^{-1}$). These regions are the same shown in Figure S6.*

**References**

Ceppi, P. and Nowack, P.: Observational evidence that cloud feedback amplifies global warming, Proc. Natl. Acad. Sci. U.S.A., 118, e2026290118, https://doi.org/10.1073/pnas.2026290118, 2021.

Fueglistaler, S.: Observational Evidence for Two Modes of Coupling Between Sea Surface Temperatures, Tropospheric Temperature Profile, and Shortwave Cloud Radiative Effect in the Tropics, Geophysical Research Letters, 46, 9890–9898, https://doi.org/10.1029/2019GL083990, 2019.

Myers, T. A., Scott, R. C., Zelinka, M. D., Klein, S. A., Norris, J. R., and Caldwell, P. M.: Observational constraints on low cloud feedback reduce uncertainty of climate sensitivity, Nat. Clim. Chang., 11, 501–507, https://doi.org/10.1038/s41558-021-01039-0, 2021.